# Allosteric mechanism for KCNE1 modulation of KCNQ1 potassium channel activation

Georg Kuenze[1,2,3]*, Carlos G Vanoye[4], Reshma R Desai[4], Sneha Adusumilli[4], Kathryn R Brewer[1,5], Hope Woods[1,2], Eli F McDonald[1,2], Charles R Sanders[1,5], Alfred L George Jr[4], Jens Meiler[1,2,3,6]

[1]Center for Structural Biology, Vanderbilt University, Nashville, United States; [2]Department of Chemistry, Vanderbilt University, Nashville, United States; [3]Institute for Drug Discovery, Leipzig University, Leipzig, Germany; [4]Department of Pharmacology, Northwestern University Feinberg School of Medicine, Chicago, United States; [5]Department of Biochemistry, Vanderbilt University, Nashville, United States; [6]Department of Pharmacology, Vanderbilt University, Nashville, United States

**Abstract** The function of the voltage-gated KCNQ1 potassium channel is regulated by co-assembly with KCNE auxiliary subunits. KCNQ1-KCNE1 channels generate the slow delayed rectifier current, $I_{Ks}$, which contributes to the repolarization phase of the cardiac action potential. A three amino acid motif (F57-T58-L59, FTL) in KCNE1 is essential for slow activation of KCNQ1-KCNE1 channels. However, how this motif interacts with KCNQ1 to control its function is unknown. Combining computational modeling with electrophysiological studies, we developed structural models of the KCNQ1-KCNE1 complex that suggest how KCNE1 controls KCNQ1 activation. The FTL motif binds at a cleft between the voltage-sensing and pore domains and appears to affect the channel gate by an allosteric mechanism. Comparison with the KCNQ1-KCNE3 channel structure suggests a common transmembrane-binding mode for different KCNEs and illuminates how specific differences in the interaction of their triplet motifs determine the profound differences in KCNQ1 functional modulation by KCNE1 versus KCNE3.

*For correspondence: georg.kuenze@gmail.com

**Competing interests:** The authors declare that no competing interests exist.

## Introduction

Voltage-gated K[+] (K$_V$) channels facilitate the movement of K[+] ions across the lipid bilayer in response to membrane depolarization and are essential for signaling in electrically excitable tissues (*Jan and Jan, 2012*). Among K$_V$ channels, KCNQ1 (K$_V$7.1, K$_V$LQT1) is special because of its wide range of physiological behaviors (*Abbott, 2014*). This versatility enables KCNQ1 to function distinctly in both excitable cells such as cardiomyocytes and in non-excitable cells such as epithelia (*Abbott, 2014*). The functional diversity of KCNQ1 is a consequence of its ability to form channel complexes with any one of five tissue-specific KCNE auxiliary proteins (KCNE1-5) (*Barhanin et al., 1996*; *Sanguinetti et al., 1996*; *Schroeder et al., 2000*; *Tinel et al., 2000*; *Teng et al., 2003*; *Angelo et al., 2002*).

KCNQ1 channels consist of four identical pore-forming subunits, each containing six membrane-spanning segments (S1-S6) and a pore loop (P loop) (*Figure 1A*). The centrally located pore domain (PD, S5-P-S6) forms the ion permeation pathway and is surrounded by four voltage-sensing domains (VSDs, S1-S4). The VSD S4 helix carries positively charged residues that trigger S4 movement upon membrane depolarization (*Bezanilla, 2000*), leading to three detectable VSD conformational states in KCNQ1 (resting, intermediate, and activated) (*Panaghie and Abbott, 2007*; *Wu et al., 2010*;

**Figure 1.** KCNQ1 channel architecture and sequence of KCNE proteins. (**A**) Topology diagram of the KCNQ1-KCNE1 channel complex. The KCNQ1 voltage-sensing (VSD, helix S1-S4), pore-forming (PD, S5–P–S6), and cytosolic domains (helix HA-HD) are colored green, blue, and gray, respectively. KCNE1 exhibits a single-span transmembrane domain (TMD) that is flanked by intra- and extracellular domains containing helical segments. (**B**) Amino acid sequence alignment of KCNE1 and KCNE3. Similar and identical amino acid residues are colored light and dark gray, respectively. The TMD region is indicated by a black box. The activation motif regions in KCNE1 and KCNE3 are highlighted in red.

*Zaydman et al., 2014*; *Taylor et al., 2020*). S4 movement is thought to exert a lateral pull on the S4-S5 linker (S4-S5L), which triggers opening of the helical S6 gate making the channel conductive (*Long et al., 2005*). KCNEs serve as β-subunits of KCNQ1 and contain a single transmembrane-spanning domain (TMD) in addition to sizeable extra- and intracellular domains (*McCrossan and Abbott, 2004*; *Figure 1A*).

Co-assembly of KCNQ1 with KCNE1 generates a channel complex that exhibits slow activation that occurs at more positive potentials and with higher conductance relative to KCNQ1 alone (*Barhanin et al., 1996*; *Sanguinetti et al., 1996*). KCNQ1-KCNE1 channels generate the slow delayed rectifier $K^+$ current ($I_{Ks}$) in the heart that contributes to the repolarization phase of the cardiac action potential. Heritable mutations in KCNQ1 and KCNE1 predispose individuals to life-threatening ventricular arrhythmia and cause type 1 and type 5 long QT syndrome (LQTS) (*Bohnen et al., 2017*), respectively. By contrast, pairing of KCNQ1 with another KCNE subunit, KCNE3, produces channels that are constitutively active over the full physiological voltage range (*Schroeder et al., 2000*).

Different mechanisms have been proposed to explain how KCNE1 modulates KCNQ1 function including alteration of S4 movement (*Nakajo and Kubo, 2007*; *Rocheleau and Kobertz, 2008*; *Osteen et al., 2010*; *Ruscic et al., 2013*; *Barro-Soria et al., 2014*), perturbation of gate opening (*Tapper and George, 2001*; *Melman et al., 2004*; *Panaghie et al., 2006*), changes in VSD-PD coupling (*Zaydman et al., 2014*; *Westhoff et al., 2019*), or a combination of these effects (*Nakajo and*

*Kubo, 2014*; *Barro-Soria et al., 2017*). However, a clear structural explanation is lacking owing to the absence of a high-resolution structure for the KCNQ1-KCNE1 complex. Previously, low-resolution spatial restraints for the KCNQ1-KCNE1 interaction were derived from disulfide crosslinking (*Chung et al., 2009*; *Wang et al., 2011*; *Chan et al., 2012*; *Wang et al., 2012*), metal ion bridging (*Tapper and George, 2001*), and site-directed mutagenesis data (*Strutz-Seebohm et al., 2011*; *Li et al., 2015*). In conjunction with computational modeling (*Strutz-Seebohm et al., 2011*; *Kang et al., 2008*; *Gofman et al., 2012*; *Xu et al., 2013*), these restraints have provided initial insight into the KCNQ1-KCNE1 channel architecture. Those models suggested that KCNE1 binds in a cleft surrounded by two VSDs and the PD, and is therefore in a location where it can simultaneously modulate S4 and the channel gate (*Kang et al., 2008*; *Gofman et al., 2012*; *Xu et al., 2013*). Additional studies have determined sites in KCNE1 that are crucial for its functional modulation of KCNQ1. Specifically, a three amino acid motif (F57-T58-L59, FTL) in the middle of the KCNE1 TMD (*Figure 1B*) was found to be necessary for induction of slow activation of KCNQ1 (*Melman et al., 2001*; *Melman et al., 2002*). Replacement of the corresponding segment in KCNE3 (T71-V72-G73, TVG) with FTL confers KCNE1-like gating properties onto the KCNQ1-KCNE3 channel (*Barro-Soria et al., 2017*; *Melman et al., 2001*; *Melman et al., 2002*). Likewise, mutation of FTL to TVG renders the KCNQ1-KCNE1 channel similar to KCNQ1-KCNE3, in that faster activation at more negative potentials is observed (*Barro-Soria et al., 2017*; *Melman et al., 2001*). How this so-called 'activation motif' determines the distinct gating properties of KCNQ1-KCNE channels is unclear. Mutations of residues in S6 (S338, F339, F340) alter the effect of mutations in the activation motif and vice versa, which was interpreted as a consequence of direct physical interaction between KCNE1 and S6 (*Panaghie et al., 2006*). However, this idea has been challenged recently, because these S6 residues reside deep within the PD in the KCNQ1 structure (*Sun and MacKinnon, 2017*; *Sun and MacKinnon, 2020*) and cysteine exchange experiments failed to confirm disulfide bond formation with any of the residues in the activation motif (*Xu et al., 2013*).

Here, we combined computational protein-protein docking, molecular dynamics, and electrophysiology to develop refined molecular models for the KCNQ1-KCNE1 complex to address the question of how the KCNE1 TMD modulates activation gating. Our results suggest that the KCNE1 FTL motif interacts with sites in S1, S4, and S5 in KCNQ1, and affects the channel gate by an allosteric network involving the S5-S6 interface. Comparison of independently constructed KCNQ1-KCNE1 models with the recently determined structures of the KCNQ1-KCNE3 complex (*Sun and MacKinnon, 2020*) shows a conserved TMD-binding mode for KCNE1 and KCNE3, but reveals specific differences in the interaction of the activation motif with the channel, consistent with the different effects of these KCNE proteins on channel gating (*Barro-Soria et al., 2017*). Our results provide more precise information on the state-specific structural requirements and specificity of KCNE subunit interactions with KCNQ1.

## Results

### Probing the spatial proximity of KCNQ1 V141 and I274 to KCNE1

For building KCNQ1-KCNE1 models, we collected residue contact restraints from previously published biophysical experiments on the KCNQ1-KCNE1 interaction: disulfide crosslinking (*Chung et al., 2009*; *Wang et al., 2011*; *Chan et al., 2012*; *Wang et al., 2012*), Cd(II)-cysteine bridging (*Tapper and George, 2001*), and double mutant cycle analysis (*Strutz-Seebohm et al., 2011*; *Li et al., 2015*). Restraints were compiled in a state-dependent manner based on the data informing whether a crosslink or mutation favored the open or closed channel state. Most restraints were available for the region N-terminal to the KCNE1 TMD (S37-A44) whereas a smaller number of restraints fell within the TMD (L45-L71). To obtain additional contact information for the KCNE1 TMD, we performed KCNQ1-KCNE1 disulfide trapping experiments and tested if KCNQ1 V141 and I274 are close to L45, V47, or L48 in KCNE1. Residues V141 and I274 are the location of two of five known KCNQ1 gain-of-function mutations within the putative KCNE1-binding region between the VSD and PD: S140G (*Chan et al., 2012*; *Chen et al., 2003a*; *Peng et al., 2017*), V141M (*Chan et al., 2012*; *Peng et al., 2017*; *Hong et al., 2005*), I274V (*Arnestad et al., 2007*), A300T (*Bianchi et al., 2000*), V307L (*Bellocq et al., 2004*). For three of those mutations (V141M, I274V, V307L) a gain-of-function phenotype is observed only when KCNE1 is present, suggesting a physical interaction.

Indeed, in cysteine exchange experiments, V141C was previously shown to crosslink with residues flanking the KCNE1 TMD on the extracellular side (*Chung et al., 2009*; *Wang et al., 2011*; *Chan et al., 2012*). We expanded upon these earlier studies and tested for disulfide bond formation between cysteines introduced at V141 or I274 in KCNQ1 and L45, V47, or L48 in KCNE1 using oxidation-state dependent electrophysiology measurements (*Figure 2*, *Figure 2—figure supplements 1–3*). L45, V47, and L48 were selected because their sidechains are oriented toward KCNQ1 in an earlier KCNQ1-KCNE1 model by *Kang et al., 2008*.

The dual mutation KCNQ1 I274C+KCNE1 L45C led to a channel that was highly conductive under reducing conditions, but displayed lower peak current under oxidizing conditions (*Figure 2A*). This result indicates that disulfide bond formation between these cysteine-substituted residues trapped the KCNQ1-KCNE1 channel in a conformation that favors the closed state. In contrast, channels formed by co-expression of KCNQ1 V141C with KCNE1 L48C exhibited smaller current amplitude under reducing conditions (*Figure 2B*), indicating that disulfide bond formation between V141C and L48C favored the open state. These results are consistent with the prediction that L45 and L48 are in

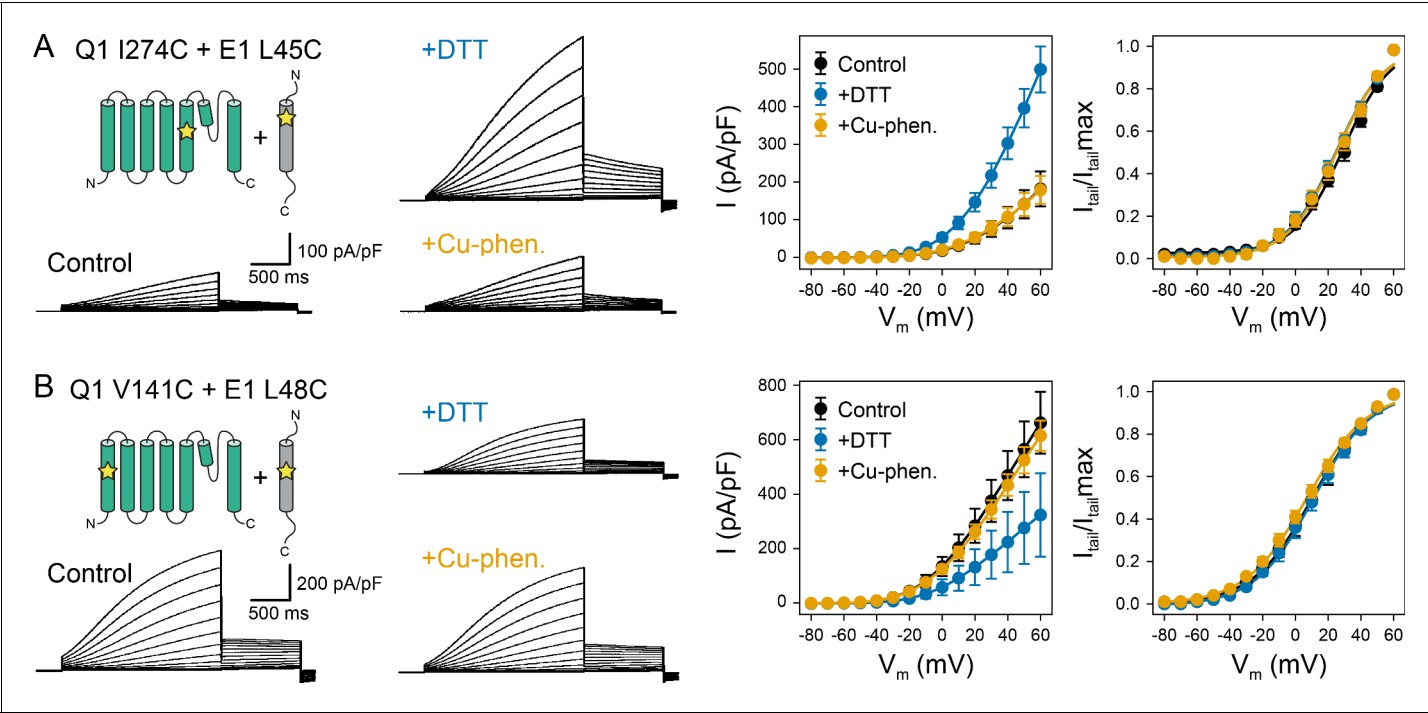

**Figure 2.** Oxidation state-dependent electrophysiology measurements indicate that KCNQ1 V141 and I274 are close to L48 and L45 in KCNE1, respectively. (**A**) Whole-cell currents (left) and average current-voltage (I–V) relationships (right) of CHO-K1 cells transiently expressing KCNQ1 I274C and KCNE1 L45C. Cells were exposed to control bath solution containing DTT or Cu-phenanthroline (Cu-phen.). (mean ± SEM, Control n = 6, DTT n = 6, Cu-phen n = 5). (**B**) Whole-cell currents (left) and average I-V relationships (right) of CHO-K1 cells expressing KCNQ1 V141C and KCNE1 L48C, which were exposed to control bath solution, DTT, or Cu-phenanthroline, respectively. (Control n = 7, DTT n = 5, Cu-phen n = 5). Solid lines represent fits with a Boltzmann function ($I_{tail}/I_{tail}max = (1-I_{Bottom}) / (1+exp[(V_{1/2app}-V)/k]) + I_{Bottom}$) and the parameters of the fit are summarized in *Supplementary file 1 – Table 1*. Control measurements of KCNQ1 WT, V141C, and I274C with and without KCNE1 under reducing (+DTT) and oxidizing (+Cu-phen.) conditions are displayed in *Figure 2—figure supplement 1* and *Figure 2—figure supplement 2*.

The online version of this article includes the following source data and figure supplement(s) for figure 2:

**Source data 1.** Excel file with numerical electrophysiology data used for *Figure 2*.

**Figure supplement 1.** Electrophysiology measurements of KCNQ1 WT, V141C, and I274C under reducing and oxidizing conditions.

**Figure supplement 1—source data 1.** Excel file with numerical electrophysiology data used for *Figure 2—figure supplement 1*.

**Figure supplement 1—source data 2.** Excel file with numerical electrophysiology data used for *Figure 2—figure supplement 2*.

**Figure supplement 1—source data 3.** Excel file with numerical electrophysiology data used for *Figure 2—figure supplement 3*.

**Figure supplement 2.** Electrophysiology measurements of KCNQ1 WT, V141C, and I274C with KCNE1 WT under reducing and oxidizing conditions.

**Figure supplement 3.** Electrophysiology measurements of KCNQ1 V141C or I274C with KCNE1 V47C or L48C under reducing and oxidizing conditions.

spatial proximity to KCNQ1 I274 and V141, respectively. The mutant KCNQ1 channels alone (*Figure 2—figure supplement 1*) or in the presence of WT KCNE1 (*Figure 2—figure supplement 2*) were insensitive to the addition of DTT or Cu-phenanthroline. Other KCNQ1-KCNE1 residue pairs tested (V141C-V47C, I274C-V47C, I274C-L48C) (*Figure 2—figure supplement 3*) showed no changes under reducing or oxidizing conditions. Together, these results show that the effect of DTT and Cu-phenanthroline is dependent on the presence of introduced cysteines at KCNQ1 residues V141, I274 and KCNE1 residues L48 and L45.

## Development of integrated structural models of the KCNQ1-KCNE1 complex

The above experimental information and the structural data collected from the literature were used as contact restraints to develop structural models for the KCNQ1-KCNE1 complex by molecular docking. Separate restraint lists for building KCNQ1-KCNE1 channel models in closed and open states were compiled (*Supplementary file 1 – Table 2 and 3*). As input for docking, we used models of human KCNQ1 (*Kuenze et al., 2019*) with the VSD and PD in resting/closed (RC) or fully activated/open (AO) conformations. Those models were previously developed based on homology modeling with the structures of *X. laevis* KCNQ1 (*Sun and MacKinnon, 2017*) and the K$_V$1.2/2.1 chimera (*Long et al., 2007*). The models have been recently confirmed by cryo-electron microscopy (EM)-determined structures of human KCNQ1 (*Sun and MacKinnon, 2020*), to which the model-predicted VSD and PD conformations are highly similar (*Figure 3—figure supplement 1*; Cα-RMSD for VSD and PD less than 1.9 Å and 2.0 Å, respectively). In the putative KCNE1-binding region used for docking, the homology models agree well with the cryo-EM structures; surface-exposed residues have a sidechain RMSD less than 2.5 Å and 4.0 Å in the RC and AO model, respectively (*Figure 3—figure supplement 1*). In addition, homology modeling provided a conformation for the VSD in the resting state, a state for which no experimental structure exists.

Using Rosetta protein-protein docking (*Gray et al., 2003*; *Gray, 2006*) and Rosetta Membrane potentials (*Yarov-Yarovoy et al., 2006*; *Barth et al., 2007*), the ensemble of ten models of the NMR-determined KCNE1 TMD structure (S37-L71) (PDB: 2K21) (*Kang et al., 2008*) was docked to the KCNQ1 models (*Kuenze et al., 2019*). We focused our structural studies on the isolated KCNE1 TMD (including a short stretch of the N-terminal TMD-flanking region (S37-A44)) because previous studies had demonstrated that this domain alone is sufficient to produce the slow activation kinetics and increased current amplitude expected for KCNQ1-KCNE1 channels (*Melman et al., 2001*). Models were generated by iterative rounds of protein-protein docking (*Figure 3—figure supplement 2*), each with a rigid-body docking phase and an all-atom flexible backbone and sidechain refinement phase, resulting in a steady optimization of the model restraint score and minimization of the Rosetta-calculated binding energy ($\Delta G_{Binding}$) for the KCNQ1-KCNE1 interaction (*Figure 3—figure supplement 3*). The most favorably scoring models of the KCNQ1-KCNE1 complex in the RC and AO conformation that exhibit the best combined experimental restraint, Rosetta $\Delta G_{Binding}$, and MolProbity scores (*Supplementary file 1 – Table 4*) are illustrated in *Figure 3*. The atomic coordinates for these models are included in the supporting material for this paper (*Supplementary files 2* and *3*) and can be obtained from PDB-Dev (PDBDEV: 00000042) (*Vallat et al., 2018*).

Within the KCNQ1-KCNE1 models, the KCNE1 TMD is bound in a cleft formed by S6 from one KCNQ1 subunit, S5, the P helix from a second subunit, and S1 and S4 from a third (*Figure 3B+C*). The N-terminal end of the KCNE1 TMD leans towards S1, S5, and S6, and its C-terminal end contacts S1 and the bottom of S4. The FTL motif resides deep in the membrane and is oriented toward KCNQ1 (*Figure 3B+C*). In the AO model, the KCNE1 TMD C-terminus forms additional interactions with the cytosolic end of S6, which is kinked toward the membrane. Those interactions are absent in the RC model, in which S6 extends more vertically into the cytoplasm. Other than these small differences, the KCNE1 TMD-binding mode within the RC and AO KCNQ1 channel is deemed similar.

The KCNQ1-KCNE1 models satisfied the experimental restraints remarkably well. Many of the restrained KCNQ1-KCNE1 Cα-Cα distances were below the upper cutoff (12 Å) employed in docking (*Figure 4*) and came close to the expected maximal Cα-Cα crosslinking distance (disulfide: 7–8 Å, cysteine-Cd(II)-cysteine: 10–11 Å) when considering dynamics in the protein model by conducting MD simulations (*Supplementary file 1 – Table 2 and 3*). Only one medium restraint violation (3.5 Å) and two large restraint violations (>5 Å) were observed in the RC model. The restraint with the medium violation involved residues KCNE1 L45 and KCNQ1 I274. It is possible that crosslinking

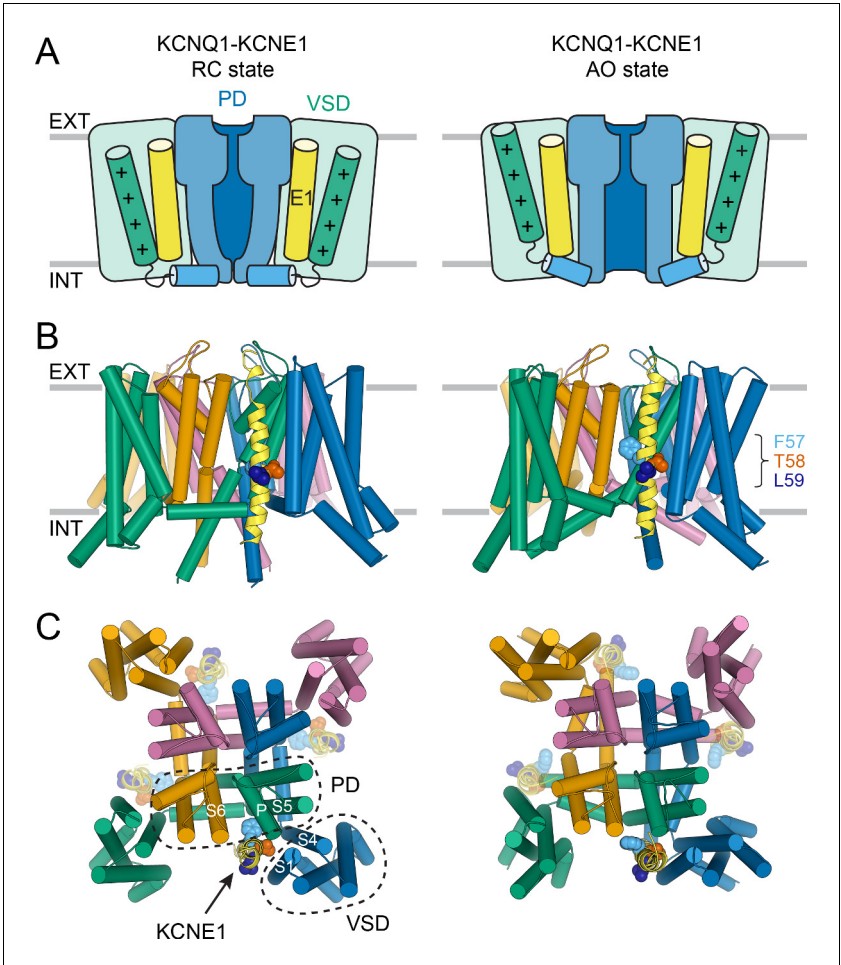

**Figure 3.** Molecular models of the KCNQ1-KCNE1 channel in RC and AO conformations. (**A**) Schematic cartoon depicting the functional states of the VSD (green box) and PD (blue box) in the KCNQ1-KCNE1 models. The S4 helix (with positive gating charges "+") and S4-S5L, which connects S4 to the PD, are shown as green and blue cylinders, respectively. KCNE1 was docked to KCNQ1 with the VSD/PD in the resting/closed (RC) or activated/open (AO) conformation. (**B**) Side view of the KCNQ1-KCNE1 docking models. KCNQ1 is represented with cylindrical helices and KCNE1 is depicted as yellow ribbon. Residues F57, T58, and L59 are drawn as spheres and colored light blue, red, and dark blue, respectively. The approximate position of the membrane bilayer is indicated by horizontal lines and the extracellular and intracellular side are labeled EXT and INT, respectively. (**C**) View of the KCNQ1-KCNE1 models from the extracellular side. KCNE1 is bound in a cleft between the VSD and PD and makes contacts to three KCNQ1 subunits. The position of the other three equivalent KCNE1-binding sites in the tetrameric KCNQ1 channel is indicated.

The online version of this article includes the following source data and figure supplement(s) for figure 3:

**Figure supplement 1.** Structural comparison of Rosetta-generated computational models of human KCNQ1, which were used for docking, with cryo-EM-determined models of human KCNQ1 (*Sun and MacKinnon, 2020*).

**Figure supplement 2.** Flowchart of the Rosetta protein-protein docking protocol for building KCNQ1-KCNE1 models.

**Figure supplement 3.** Rosetta binding energy ($\Delta G_{Binding}$) versus interface root-mean-square deviation (RMSD) plots of KCNQ1-KCNE1 docking models.

**Figure supplement 3—source data 1.** Excel file with numerical data used for the energy-vs-RMSD plots in panel A of *Figure 3—figure supplement 3*.

**Figure supplement 3—source data 2.** Excel file with numerical data used for the energy-vs-RMSD plots in panel B of *Figure 3—figure supplement 3*.

**Figure supplement 4.** Control docking calculations for KCNQ1-KCNE1 and KCNQ1-KCNE3 complexes using the Rosetta and cryo-EM models of the activated/open state structure.

*Figure 3 continued on next page*

*Figure 3 continued*

**Figure supplement 4—source data 1.** Excel file with numerical data used to make the energy-vs-I-RMSD plots in panel A of *Figure 3—figure supplement 4*.

**Figure supplement 4—source data 2.** Excel file with numerical data used to make the energy-vs-I-RMSD plots in panel B of *Figure 3—figure supplement 4*.

**Figure supplement 4—source data 3.** Excel file with numerical data used to make the energy-vs-I-RMSD plots in panel C of *Figure 3—figure supplement 4*.

---

between these cysteine-substituted sites slightly perturbed the KCNQ1-KCNE1 structure leading to a low-conductance closed-like state, which could explain why in the WT channel model these residue sidechains have suboptimal geometry for disulfide bond formation. The two largest violations involved pairs of residues originally believed to be in direct contact based on results of double mutant cycle experiments: KCNE1 T58–KCNQ1 F340 (*Strutz-Seebohm et al., 2011*) and KCNE1 Y65–KCNQ1 A344 (*Li et al., 2015*). In our KCNQ1 RC channel model (*Kuenze et al., 2019*) as well as in the experimental KCNQ1 structure (*Sun and MacKinnon, 2020*), F340 and A344 are deeply buried within the PD and inaccessible to KCNE1 T58 and Y65. Satisfying either of the two restraints would require hard-to-rationalize conformational changes in KCNE1 and/or KCNQ1. Moreover, Xu

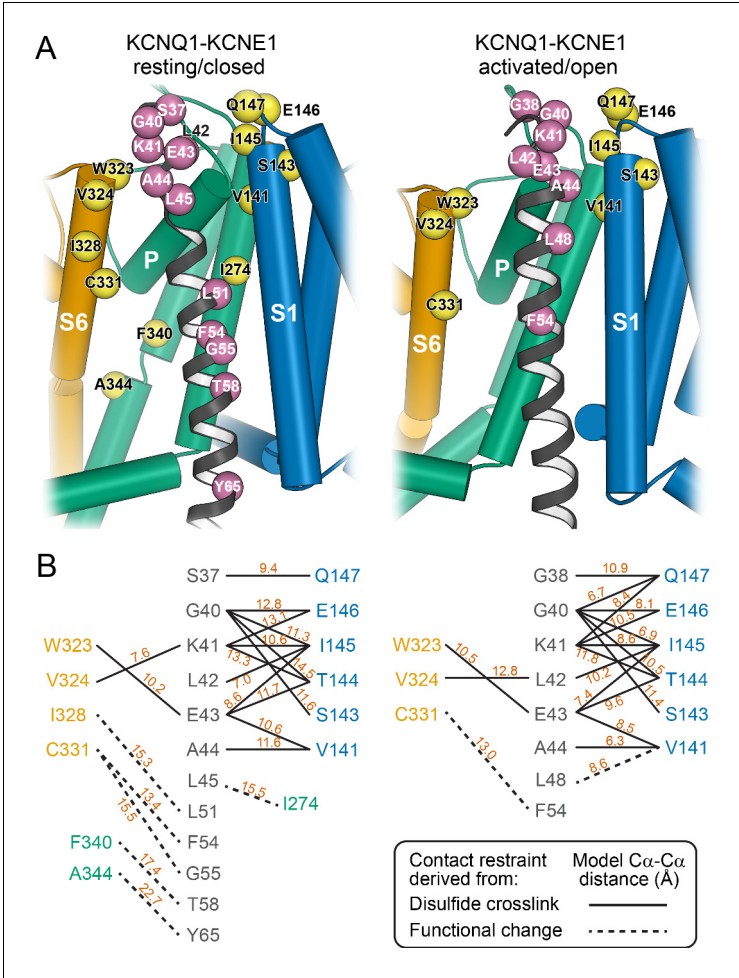

**Figure 4.** Mapping of experimental distance restraint sites onto the KCNQ1-KCNE1 models. (**A**) Interaction of KCNE1 with the KCNQ1 RC and AO model in the transmembrane region. Residues whose distance was restrained in docking are indicated as spheres. (**B**) Restrained residue pairs and their Cα-Cα distances (in Å) in the KCNQ1-KCNE1 RC and AO models.

and coworkers (*Xu et al., 2013*) confirmed previously that the FTL motif residues are unable to form disulfide bonds with S338, F339, or F340 in KCNQ1, suggesting that energetic coupling between those KCNQ1-KCNE1 residue pairs is mediated by allosteric networks rather than a direct interaction.

To assure the robustness of our structure prediction protocol, control docking calculations were performed with the cryo-EM-determined AO state structure of human KCNQ1 (*Sun and MacKinnon, 2020*), which became available only after our KCNQ1-KCNE1 models were completed. These control calculations arrived at a model that was very similar to the one developed by docking KCNE1 to the Rosetta homology model of KCNQ1 (interface RMSD (I-RMSD) = 3.5 Å, *Figure 3—figure supplement 4A*). Additional control calculations were carried out with KCNE3, starting either with the cryo-EM-determined or Rosetta-predicted KCNQ1 model. Guided by a set of published experimental restraints for the KCNQ1-KCNE3 complex (*Kroncke et al., 2016*), this procedure was able to reproduce the experimental KCNE3 binding pose with an accuracy of I-RMSD = 2.5 Å or 4.2 Å, respectively, for these two structures (*Figure 3—figure supplement 4B and C*).

In summary, using iterative protein-protein docking and model filtering with experimental restraints, we extensively probed the KCNQ1-KCNE1 interaction and developed molecular models of the channel complex that favorably agree with the available experimental data. These models were tested by the subsequent experimental and MD analysis.

## Experimental validation of the KCNE1-binding site of KCNQ1

To gain further mechanistic insight into how KCNE1 and KCNQ1 interact, we analyzed the location and degree of their contacts by performing MD simulations of our structural models (*Figure 5A and B*, *Figure 5—figure supplement 1*) coupled with site-directed mutagenesis. For MD simulations, KCNQ1-KCNE1 models were prepared with a stoichiometry of 4:2 KCNQ1:KCNE1 subunits. This appears to be the predominant stoichiometry on the surface of mammalian cells (*Plant et al., 2014*), although the possibility for multiple ratios ranging from 4:1 to 4:4 has been discussed (*Morin and Kobertz, 2008*; *Nakajo et al., 2010*; *Murray et al., 2016*). In our modeling procedure, we did not expect to find changes in the KCNE1 interaction mode for different KCNQ1: KCNE1 ratios, because we first docked one KCNE1 molecule to tetrameric KCNQ1 and subsequently created 4:2 and 4:4 complexes by imposing C2 or C4 symmetry, respectively. In our final MD analysis, we focused on the 4:2 stoichiometry and observed no significant changes in the interaction mode between the two KCNE1 subunits and with respect to the model obtained by Rosetta docking.

The first residues in the KCNE1 TMD and TMD-flanking region (S37-G50) interact with S1, the S1-S2 loop, the P helix, and S6 in both the RC and AO channel models (*Videos 1* and *2*, *Supplementary file 1 – Table 5*). Among those residues, the largest number of contacts with KCNE1 is made by W323 at the N-terminal end of S6 (*Figure 5C*, left panel). Mutations of W323 to Ala and Leu led to KCNQ1-KCNE1 channels with faster activation (*Figure 5—figure supplement 2A*) and a significantly hyperpolarized activation curve compared to the WT channel (change in 'apparent' activation $V_{1/2}$ (see Materials and methods): $\Delta V_{1/2app,W323A}$ = -6.4 mV, $\Delta V_{1/2app,W323L}$ = -11.3 mV) (*Figure 5D+E*). Mutations of other residues in S6 (V324, V334 [*Nakajo et al., 2011*]) and in the nearby P helix (A300 [*Bianchi et al., 2000*], V307 [*Bellocq et al., 2004*]) also caused channel opening at more negative voltages, likely via destabilization of the closed state. This region has been implicated with the positive G(V) shift by KCNE1 (*Nakajo et al., 2011*). Mutation of W323 to Phe resulted in a WT-like channel ($\Delta V_{1/2app,W323F}$ = 0.5 mV) suggesting that an aromatic or large hydrophobic moiety is an important structural component required for interaction of site 323 with KCNE1. Our structural model shows that the indole ring of W323 caps the sidechain of KCNE1 Y46, which is tucked in between S6 and the P helix (*Figure 5C*).

The middle part of the KCNE1 TMD (L51-Y65) interacts with S1 and S5 in both the RC and AO channel models (*Figure 5A and B*, *Videos 1* and *2*, *Supplementary file 1 – Table 5*). KCNE1 interacts in slightly different ways with the C-terminal end of S4 in the RC and AO models owing to the movement of S4 when the VSD becomes activated (*Video 3*). Among the KCNQ1 residues interacting with the middle part of the KCNE1 TMD, Y267 in S5 had the largest number of MD contacts with KCNE1. Our structural models suggest Y267 engages in an H-bond contact with the sidechain of T58 (*Figure 5C*, middle panel). We therefore used electrophysiology to experimentally confirm an interaction with KCNE1. Substitution of Y267 with Phe, which maintains the aromatic sidechain

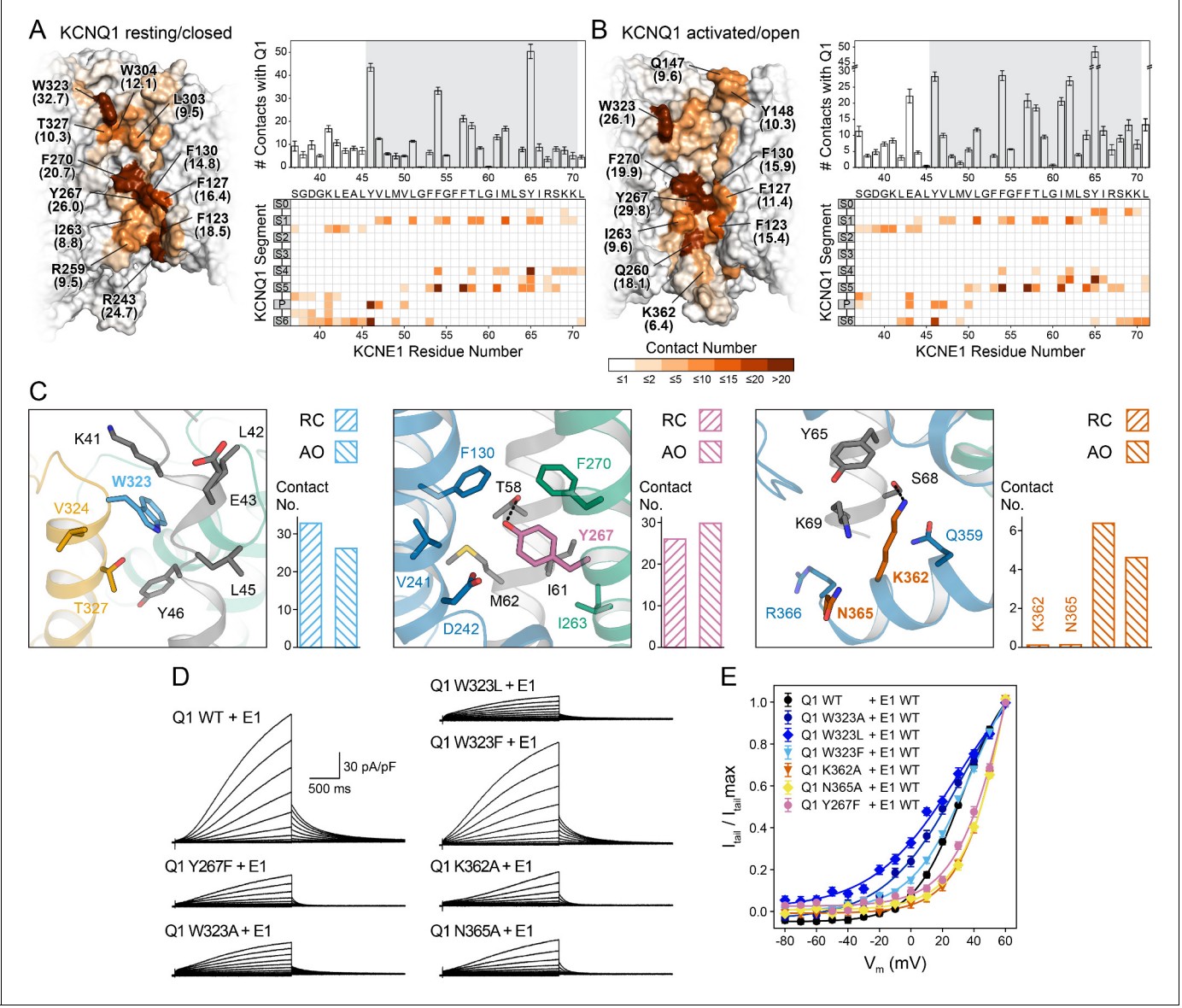

**Figure 5.** Computational detection and experimental validation of the KCNE1-binding site of KCNQ1. (**A**) Left: Surface representation of the KCNE1 binding cleft in the KCNQ1 RC model. Residues are colored by their average MD contact number with KCNE1 (indicated in parentheses). Right: Matrix of KCNQ1-KCNE1 contacts (bottom) and histogram of the number of intermolecular contacts for KCNE1 (top) (mean ± SD). (**B**) Left: Surface representation of the KCNE1 binding site in the KCNQ1 AO model with residues colored by their average MD contact number with KCNE1. Right: Matrix and histogram of the number of intermolecular contacts for KCNE1 (mean ± SD). (**C**) Interaction of KCNQ1 with the upper, middle, and lower part of the KCNE1 TMD. Three selected sites in KCNQ1 and their neighboring residues in KCNQ1 and KCNE1 are displayed: left – W323, middle – Y267, right – K362+N365. Residue sidechains are drawn as sticks and potential H-bond contacts are indicated by dashed lines. Histograms of the average MD contact number with KCNE1 for the selected residues in the KCNQ1 RC and AO model are shown next to the structural models. (**D**) Whole-cell currents of CHO-K1 cells stably expressing KCNE1 and transfected with KCNQ1 WT or mutant cDNA. (**E**) Normalized activation curves for currents recorded from cells expressing KCNQ1 WT or mutants. (mean ± SEM, WT n = 45, W323A n = 22, W323L n = 25, W323F n = 58, Y267F n = 22, K362A n = 31, N365A n = 24).

The online version of this article includes the following source data and figure supplement(s) for figure 5:

**Source data 1.** Excel file with numerical data used for panels A, B, and E of *Figure 5*.

**Figure supplement 1.** MD simulations for the KCNQ1-KCNE1 RC and AO channel models.

**Figure supplement 1—source data 1.** Excel file with numerical MD simulation data used for panels C, D, and E of *Figure 5—figure supplement 1*.

**Figure supplement 2.** Activation ($\tau_{act}$) and deactivation times ($\tau_{deact}$) of WT and mutant KCNQ1-KCNE1 channels.

**Figure supplement 3.** Cartoon model for the interaction of KCNQ1 residues H363 and I368 with KCNE1 residues H73, S74, and D76.

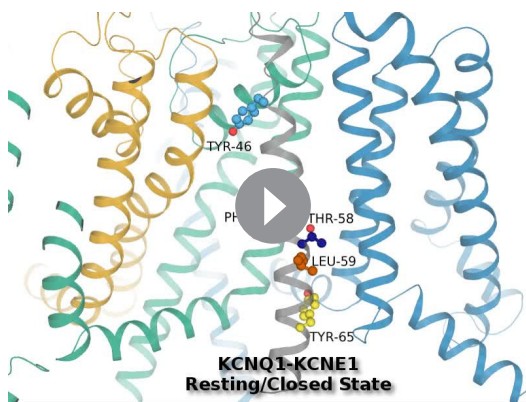

**Video 1.** Animation of KCNQ1-KCNE1 interaction sites in the KCNQ1-KCNE1 RC channel model. https://elifesciences.org/articles/57680#video1

character but lacks a hydroxyl group, resulted in a significant alteration of the voltage-dependence of KCNQ1-KCNE1 activation ($\Delta V_{1/2app, Y267F}$ = 13.5 mV) (*Figure 5D+E*). We also noted that channel activation was previously shown to be altered by Ala mutations at Y267 and other residues in S5, mostly L266 and F270 (*Strutz-Seebohm et al., 2011*), which underscores the importance of the S5-KCNE1 interface.

At its cytosol-proximal end, the KCNE1 TMD interacts with the S0-S1 loop, S4, and, in the AO channel model, additionally with the loop connecting S4 and S4-S5L and with the C-terminal end of S6 (*Figure 5C* right panel, *Videos 1* and *2*). The latter interaction may specifically contribute to the stability of the open state by reducing the dynamics of S6 and locking the S6 gate open. To test this hypothesis, we introduced Ala mutations at K362 and N365 in S6, and determined the effect on the voltage-dependence of KCNQ1 activation. Both mutations resulted in significantly depolarized activation $V_{1/2app}$ ($\Delta V_{1/2app,K362A}$ = 14.7 mV, $\Delta V_{1/2app,N365A}$ = 14.7 mV) (*Figure 5D+E*) and significantly faster deactivation (*Figure 5—figure supplement 2A*), indicating that these mutant channels required more energy to open. A proximity between the C-terminal ends of the KCNE1 TMD and S6 is supported by the results of cysteine-crosslinking experiments (*Lvov et al., 2010*), which showed that H363C in KCNQ1 formed disulfide bonds with H73C, S74C, and D76C in KCNE1. While these residues were not included in our docking because of their location in a flexible linker region, they are close to the last KCNE1 residue in our models. Extending the models by five additional KCNE1 residues can bring H73, S74, and D76 in contact with H363 (*Figure 5—figure supplement 3*). Furthermore, using double mutant cycle analysis, *Chen et al., 2020* recently suggested that H73, S74, and D76 can interact with another residue on S6, I368. This result can also be explained by our structural models (*Figure 5—figure supplement 3*). Interestingly, the sequence R360-Q361-K362-H363 has been observed to undergo a major conformational change during KCNQ1 gating (*Sun and MacKinnon, 2020*). This structural change has been associated with PIP2 binding (*Sun and MacKinnon, 2020*). Analysis of our KCNQ1-KCNE1 model suggests interaction

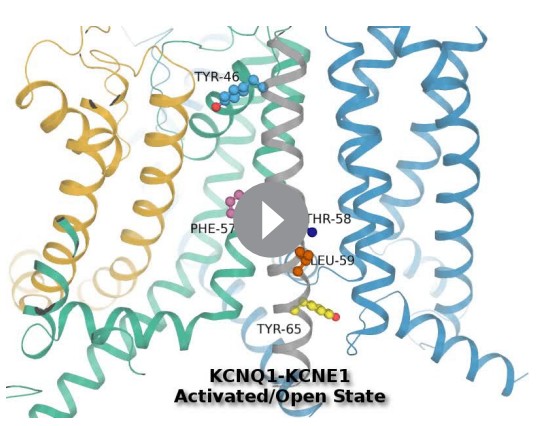

**Video 2.** Animation of KCNQ1-KCNE1 interaction sites in the KCNQ1-KCNE1 AO channel model. https://elifesciences.org/articles/57680#video2

**Video 3.** Morph between KCNQ1-KCNE1 RC and AO models illustrating S6 helix kinking at the PAG motif during channel opening and the location of the KCNE1 FTL motif. https://elifesciences.org/articles/57680#video3

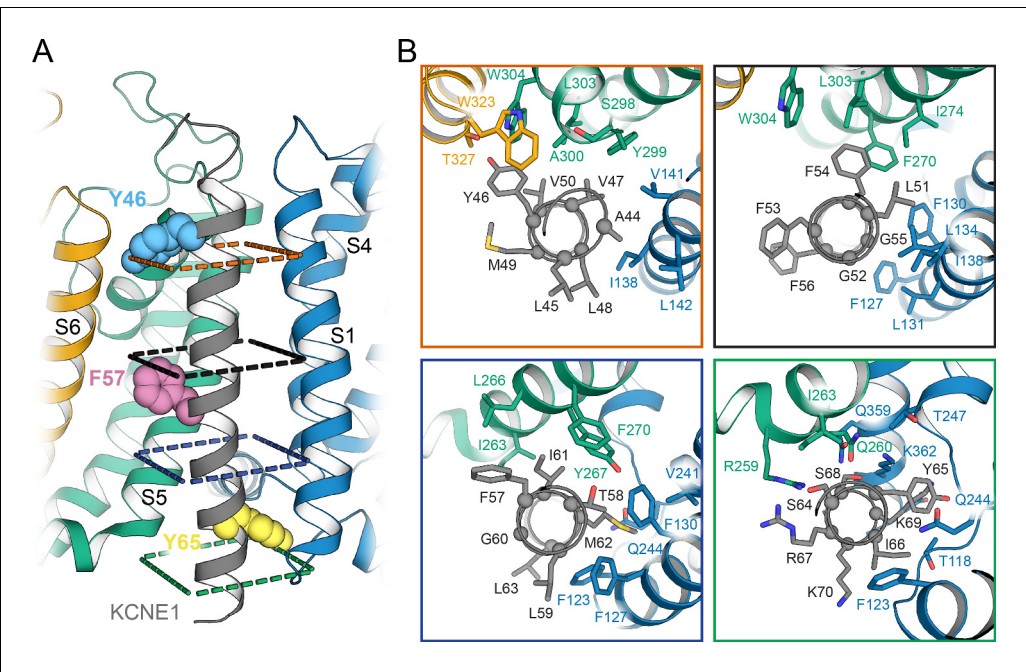

**Figure 6.** Orientation of the KCNE1 TMD in KCNQ1. (**A**) Cartoon representation of the KCNE1 TMD and its surrounding helical segments in the KCNQ1 AO model. The KCNQ1-KCNE1 RC model is shown in *Figure 6—figure supplement 1*. Residues Y46, F57, and Y65, at which mutation to Ala or Leu led to a significant change in $V_{1/2app}$ of KCNQ1 activation (*Figure 7*), are drawn as spheres. (**B**) View of the KCNE1-KCNQ1 interface from the extracellular side at planes indicated in (**A**). KCNQ1-KCNE1 residue interactions in the RC and AO model are also shown in *Videos 1* and *2*, respectively.

The online version of this article includes the following figure supplement(s) for figure 6:

**Figure supplement 1.** Orientation of the KCNE1 TMD in the KCNQ1 RC model.

---

with KCNE1 may play an additional stabilizing role to this channel-lipid interaction.

Taken together, these results have identified several important interactions in the transmembrane KCNE1-binding site of KCNQ1. We next focused on identifying and validating the KCNE1 sites that interacted with KCNQ1 in the MD simulations.

## Experimental validation of KCNE1 residues interacting with KCNQ1

The KCNE1 residues observed in MD simulations to make many contacts with KCNQ1 include Y46, V47, L51, F54, F57, T58, I61, M62, and Y65 in both the RC and AO channel models (*Figure 5A and B*). Additionally, in the AO model, S64, I66, and K69 interact with KCNQ1 S6 (*Figure 5C*, right panel). Residues Y46 and Y65, which border the KCNE1 TMD on the extra- and intracellular side, respectively, make the most contacts (*Figure 6*, *Figure 6—figure supplement 1*). Y46 is tucked in between S6 and the P helix. Y65 occupies the space below S4 and the loop connecting S4 and S4-S5L. Interactions at those two outermost points appear to anchor KCNE1 in its binding cleft and define its helical orientation (*Figure 6A*, *Videos 1* and *2*). The FTL residues are in the interface with KCNQ1; F57 is packed against S5, T58 is deeply buried between S1, S4, and S5, and L59 directly interacts with S1 (*Figure 6B*, *Videos 1* and *2*).

To validate this model-predicted binding mode, we correlated the pattern of KCNQ1-contacting positions in KCNE1 with site-directed mutagenesis data for KCNE1 (*Figure 7*). Exhaustive mutational scans of the KCNE1 TMD with Cys (*Wang et al., 2012*) or Trp and Asn (*Chen and Goldstein, 2007*) were previously reported. In addition, we tested Ala mutations at selected positions across the KCNE1 TMD and studied the mutational effects on voltage-dependent activation and gating kinetics of the resulting KCNQ1-KCNE1 channels (*Figure 7A*; *Supplementary file 1 – Table 6*). Aromatic residues were also mutated to Leu, and residues Y46 and Y65 were additionally changed to Phe.

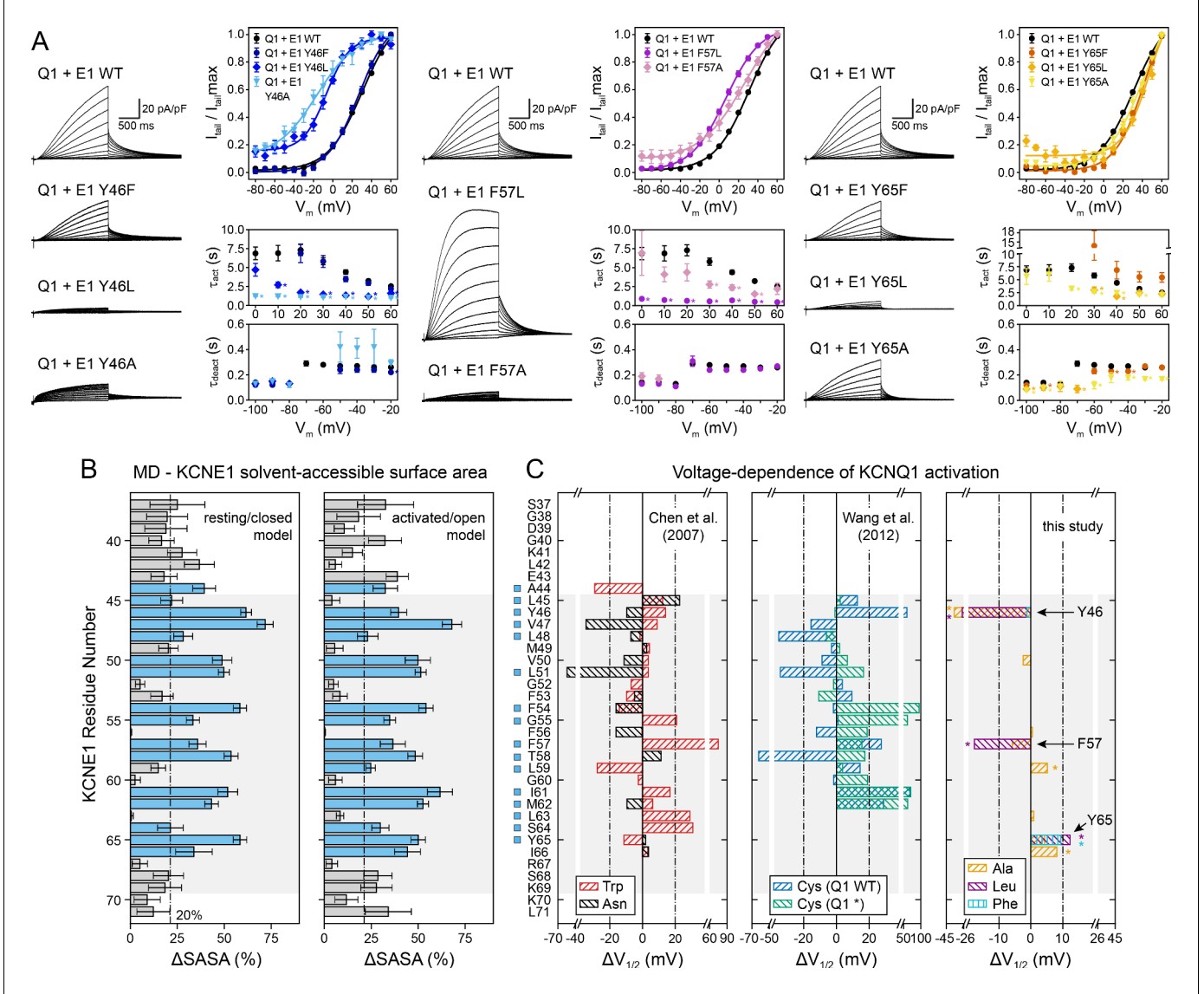

**Figure 7.** Experimental validation of KCNE1 TMD residues interacting with KCNQ1. (**A**) Whole-cell currents measured from CHO-K1 cells transiently expressing KCNQ1 with KCNE1 WT or KCNE1 variants carrying mutations at residues Y46, F57, or Y65, respectively. Normalized activation curves are shown next to the currents (mean ± SEM, WT n = 270, Y46F n = 28, Y46L n = 14, Y46A n = 14, F57L n = 58, F57A n = 14, Y65F n = 23, Y65L n = 23, Y65A n = 18). Solid lines represent fits with a Boltzmann function ($I_{tail}/I_{tail}$max = (1-$I_{Bottom}$) / (1+exp[($V_{1/2app}$-$V$)/$k$]) + $I_{Bottom}$) with the parameters of the fit summarized in **Supplementary file 1 – Table 6**. Activation time constants ($\tau_{act}$) and deactivation time constants ($\tau_{deact}$) from fits to currents at each potential (WT n = 56–293, Y46F n = 8–36, Y46L n = 15–32, Y46A n = 22–30, F57L n = 58–66, F57A n = 3–17, Y65F n = 4–24, Y65L n = 17–57, Y65A n = 10–29). Time constants significantly different from those of WT KCNQ1+KCNE1 are indicated (*p<0.001, Student's t-test). (**B**) Change in residual solvent-accessible surface area ($\Delta$SASA) between KCNE1 alone and KCNE1+KCNQ1 calculated from MD simulations of the KCNQ1-KCNE1 RC and AO model (mean ± SD). $\Delta$SASA values > 20% are shown with blue bars and indicate that a residue is part of the KCNQ1-KCNE1 interface. The approximate region of the KCNE1 TMD is indicated in gray. (**C**) Change in voltage-dependence of KCNQ1 activation by mutations in KCNE1. $\Delta V_{1/2}$ values of Trp and Asn mutants were previously reported by **Chen and Goldstein, 2007**, and those of Cys mutants are from **Wang et al., 2012**. In the latter case, experiments were performed with WT or Cys-less KCNQ1 (Q1*). Positions where mutations led to a significant change in $V_{1/2}$ ($|V_{1/2}|$>20 mV for KCNE1 expressed in oocytes in previous studies (**Wang et al., 2012**; **Chen and Goldstein, 2007**), $|V_{1/2}|$>10 mV for KCNE1 expressed in CHO-K1 cells in this study) are indicated (■). (mean, *p<0.001, Student's t-test).

The online version of this article includes the following source data and figure supplement(s) for figure 7:

**Source data 1.** Excel file with numerical data used for panels A-C in **Figure 7**.

**Figure supplement 1.** Comparison between the model-predicted KCNE1 TMD orientation and the pattern of $V_{1/2}$ changes owing to mutations in KCNE1.

**Figure supplement 1—source data 1.** Excel file with numerical data used to make panels B-D of **Figure 7—figure supplement 1**.

We defined KCNQ1-contacting positions in KCNE1 by calculating the relative change in solvent-accessible surface area ($\Delta$SASA) owing to KCNQ1-KCNE1 binding at every position in KCNE1 (*Figure 7B*). Residues with >20% $\Delta$SASA are partially or fully buried in the KCNQ1-KCNE1 interface and expected to be more sensitive to mutation. We observed a fairly good match between the pattern of KCNE1 interface positions (in both the RC and AO model) and the location of high-impact mutation sites (according to a $|\Delta V_{1/2}|$ threshold criterion; see legend to *Figure 7*). 87% (14/16) of the high impact mutation sites in KCNE1 were located at the interface with KCNQ1 ($\Delta$SASA >20%), and 71% (5/7) of the positions with low impact on channel activation were solvent-exposed ($\Delta$SASA <20%) (*Figure 7C*). Thus, there is a clear dependence between KCNQ1-contact sites and high-impact mutation sites in KCNE1 for the orientation proposed by the structural models ($p<0.05$, $\chi^2$-test). However, no significant correlation was found when we simulated other hypothetical orientations for KCNE1 by rotation around its helical screw axis (*Figure 7—figure supplement 1*). This result and the following in-depth mutational analysis of selected KCNE1 residues support the model-predicted binding mode for KCNE1.

Mutation of Y46 to Ala and Leu produced channels that were activated at more negative potentials ($\Delta V_{1/2app,Y46A}$ = -36.3 mV, $\Delta V_{1/2app,Y46L}$ = -27.7 mV) and had faster activation kinetics (*Figure 7A*, left panel). These data suggest that mutating Y46 destabilized the closed state. This observation resembles the mutational phenotype for KCNQ1 W323 (*Figure 5E*). In our structural model, W323 and Y46 are in direct contact, which can explain why mutations at either position perturb KCNQ1-KCNE1 function in a similar manner. Simultaneous substitution of both W323 and Y46 with Ala led to a channel with almost complete loss of current, which prohibited us from confirming this interaction by double mutant cycle analysis. Substituting W323 or Y46 with Phe, however, maintained WT-like channel properties (*Figures 5E* and *7A*), which suggests that an aromatic or large hydrophobic sidechain is a sufficient structural property required for interaction between these residues. In this regard, it is worth mentioning that the sidechain properties at Y46 also influence ion conduction through the KCNQ1-KCNE1 pore, as demonstrated by *Xu et al., 2013*. Small amino acids (Gly, Cys) and positively charged sidechain modifications (Cys-MTSET) increased the conductance of $Cs^+$ relative to $K^+$, whereas aromatic amino acids (Phe, Trp) did not significantly change the Cs/K conductance ratio compared to WT KCNE1. The position of Y46 in our structural models seems well suited to control this effect, and sidechain volume may influence the steric pressure on the nearby P helix to increase or restrict the conductance of larger $Cs^+$ ions through the selectivity filter.

Mutations at F57 and Y65 also had a significant impact on KCNQ1-KCNE1 function (*Figure 7A*). Mutations F57A and F57L produced channels that opened at more negative voltages ($\Delta V_{1/2app,F57A}$ = -5.8 mV, $\Delta V_{1/2app,F57L}$ = -17.7 mV) with faster kinetics (*Supplementary file 1 – Table 6*). Mutations at Y65 led to a shift of $V_{1/2app}$ to more positive voltages ($\Delta V_{1/2app,Y65A}$ = 4.3 mV, $\Delta V_{1/2app,Y65L}$ = 12.3 mV, $\Delta V_{1/2app,Y65F}$ = 9.4 mV), and Y65L and Y65A showed faster activation and deactivation kinetics (*Figure 7A*, right panel). This is consistent with the observation of Y65 forming sidechain packing and hydrogen bond interactions with multiple KCNQ1 residues in the MD simulations. Mutation I66A, while failing to meet our threshold criterion ($|\Delta V_{1/2app}|$ >10 mV) to be considered a high impact mutation site, still induced a significant change in activation voltage-dependence compared to WT ($\Delta V_{1/2app,I66A}$ = 8.2 mV, $p<0.001$), consistent with the interface location of this residue. In contrast, mutations at F56 and L63 failed to produce significant functional perturbations, as expected from their lipid-exposed positions.

Taken together, these results have identified several KCNE1 residues that interact with KCNQ1. We next focused on the interactions of the KCNE1 FTL motif and compared them with those of the TVG motif in KCNE3.

## Comparison of the KCNE1 TMD-binding mode with that of KCNE3

KCNE1 FTL (F57-T58-L59) is essential to induce slow activation of KCNQ1 (*Melman et al., 2001*; *Melman et al., 2002*). A hydroxylated amino acid at the middle position of this motif was previously found to be necessary for this effect (*Melman et al., 2002*). Replacement of FTL with TVG from KCNE3 shifts the G(V) curve of KCNQ1-KCNE1 channels towards that of KCNQ1-KCNE3 and removes KCNE1-specific effects on the gate and S4 movement (*Barro-Soria et al., 2017*). The proposed mechanism responsible for this effect is direct binding of FTL to the PD in KCNQ1 (*Melman et al., 2004*; *Panaghie et al., 2006*), and changed binding upon mutation to TVG. However, later studies have failed to confirm a direct interaction (*Xu et al., 2013*). Our refined models of

the KCNQ1-KCNE1 complex together with recently released structures of the KCNQ1-KCNE3 complex (*Sun and MacKinnon, 2020*) provide new insight into the binding and mode of action of the activation motifs from KCNE1 and KCNE3.

The TMDs of KCNE1 and KCNE3 bind in the same location in KCNQ1 in both open and closed states and share the same overall orientation (*Figure 8*). When superimposing the TM segments of KCNQ1 from both complexes with each other, KCNE1 and KCNE3 deviate by a Cα-RMSD of less than 3.1 Å (RC model, *Figure 8A*) and 5.9 Å (AO model, *Figure 8B*), respectively, along their TM region. Small observable differences between KCNE1 and KCNE3, such as the degree of helix curvature, are expected and are likely related to different intrinsic conformational properties of the TMDs of KCNE1 and KCNE3. Previous NMR (*Kang et al., 2008*; *Kroncke et al., 2016*) and EPR (*Sahu et al., 2014*) studies revealed that the TMDs of unbound KCNE1 and KCNE3 are curved and that the degree of helix curvature can vary. Importantly, despite these small differences, we found that homologous residues in KCNE1 and KCNE3 occupy the same spatial position and point in the same direction toward KCNQ1. For instance, KCNE1 Y46 and Y65, and their corresponding residues in KCNE3, Y60 and Y79, make similar interactions with KCNQ1 (*Figure 8C+E*). The FTL and TVG motifs also share a common binding site, between S1 and S4 from one subunit and S5 from a second subunit, but make different specific interactions with KCNQ1 (*Figure 8D*). Most strikingly, we noticed an H-bond between KCNE1 T58 and KCNQ1 Y267 in the MD simulations (*Figure 8F*), an observation that offers an explanation for why a hydroxylated amino acid in the middle of the activation motif is required for KCNE1 function. In order to test if Y267 and T58 are interacting, we performed a double mutant cycle experiment by substituting Y267 with Phe and T58 with Val, either separately or in combination, and determined the changes in the activation energy of the resulting channel complexes (*Figure 8—figure supplement 1*). The energy changes for KCNQ1 Y267F–KCNE1 ($\Delta G$ = 0.66 kcal/mol), KCNQ1–KCNE1 T58V ($\Delta G$ = −0.36 kcal/mol) and KCNQ1 Y267F–KCNE1 T58V ($\Delta G$ = 0.90 kcal/mol) were not additive, however, the net energy change ($|\Delta\Delta G|$ = 0.60 kcal/mol) was smaller than 1.0 kcal/mol, which is commonly used as lower cutoff to identify two residues as interacting. Thus, we were not able to experimentally confirm an interaction between Y267 and T58. However, we note that the free energy changes at T58 were previously observed to have a pronounced sidechain volume dependency (*Strutz-Seebohm et al., 2011*) and that substitutions to amino acids involving a more drastic change in sidechain size could reveal a stronger energetic coupling between Y267 and T58 than determined in this work. In the KCNQ1-KCNE3 complex, V72 and Y267 are not interacting. The Y267 sidechain is oriented differently and is in H-bond distance to D242 (or M238) on S4 (*Figure 8D*). Different interactions are also observed for the first and third motif residues, F57 (T71) and L59 (G73), which are bulkier in KCNE1, probably causing different steric effects on the surrounding residues in KCNQ1. Thus, our structural analysis of the KCNQ1-KCNE1 and KCNQ1-KCNE3 channel complexes suggests that the different functions of KCNE1 and KCNE3 are the consequence of distinct interactions involving their activation motif sites, as discussed below, while interactions with the top and bottom of the TMD help to preserve the same overall KCNE TMD-binding mode.

## Discussion

How KCNE subunits modify KCNQ1 function in such profoundly different ways is a longstanding topic of investigation. Our KCNQ1-KCNE1 models and subsequent comparative analysis with structures of the KCNQ1-KCNE3 channel aim to address two questions: How does the KCNE1 FTL motif interact with KCNQ1 to control KCNQ1 activation gating? And, what can be concluded about the TMD-binding mode for both KCNE1 and KCNE3 and the mechanism underlying the different impact on channel activation by KCNE3?

We found that KCNE1 FTL binds in a cleft between the KCNQ1 S1 (F127, F130), S4 (V241), and S5 (I263, L266, Y267 and F270) in both the RC and AO conformation (*Figure 8D*, *Figure 9A and B*). This binding mode places the activation motif in proximity to the conserved PAG (P343-A344-G345) motif in S6 – a segment that undergoes critical conformational changes during channel gating. Gating occurs as a consequence of S6 bending at the PAG motif, which causes S6 to swing away from the channel axis, opening the cytosolic gate. While the FTL residues are not in direct contact with S6, our structural model suggests that they can affect the nearby PAG motif through the mediation of S5 (*Figure 9D*, *Video 3*). Alanine mutational scanning of S5 previously showed that the $V_{1/2}$ shift

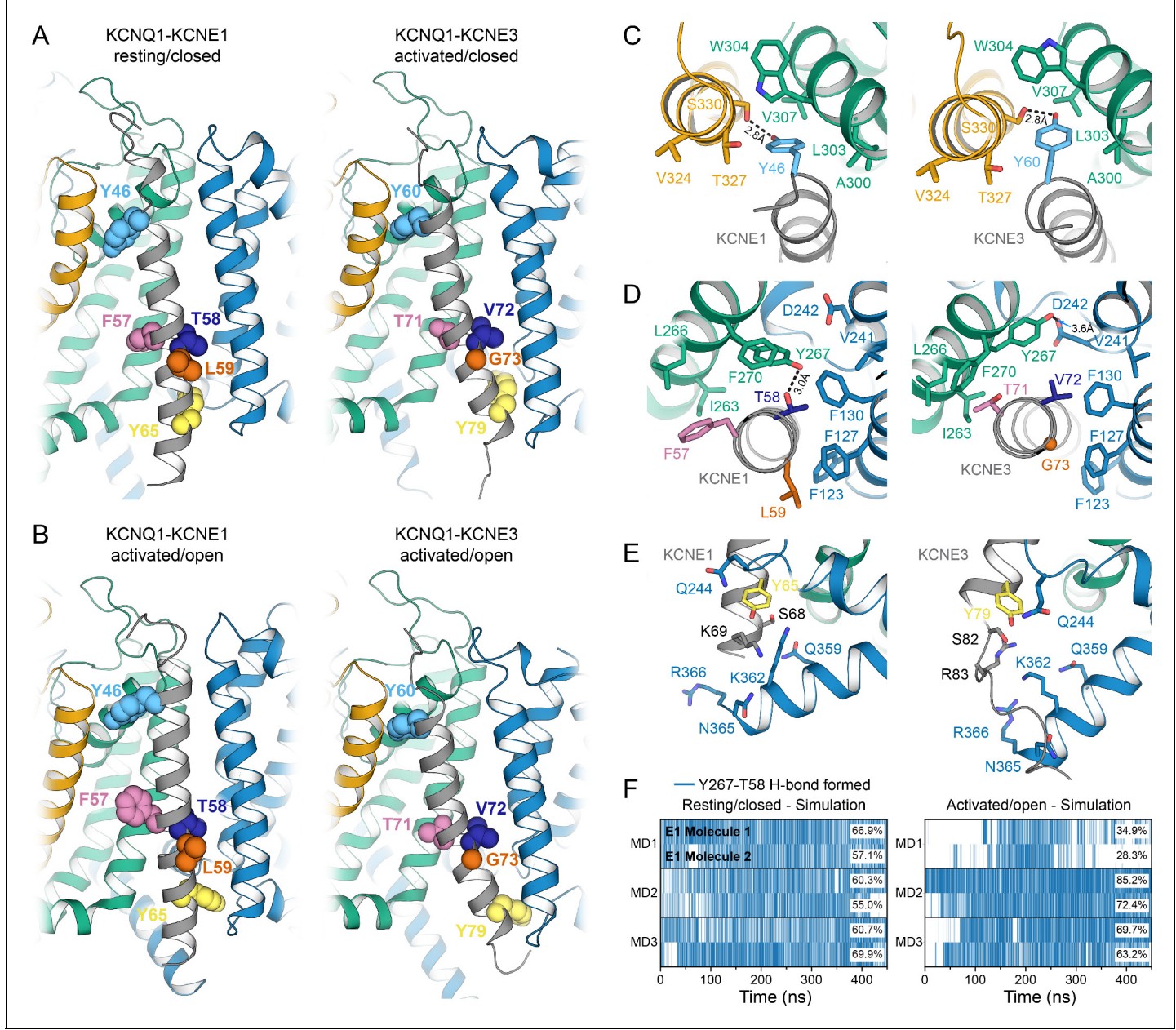

**Figure 8.** Comparison of KCNQ1-KCNE1 models with structures of the KCNQ1-KCNE3 complex. (**A**) Left: KCNE1 model bound to KCNQ1 in the RC conformation. Right: Experimental structure of KCNE3 (*Sun and MacKinnon, 2020*) bound to KCNQ1 in a decoupled state with an activated VSD and a closed PD (PDB: 6V00). KCNE1 residues Y46, F57, T58, L59, Y65, and the corresponding residues in KCNE3 are shown in spheres. (**B**) Left: KCNE1 model in complex with KCNQ1 in the AO conformation. Right: Experimental structure of KCNE3 (*Sun and MacKinnon, 2020*) bound to KCNQ1 with an activated VSD and an open PD (PDB: 6V01). (**C**) Residue neighborhood around Y46 in KCNE1 and its homologous residue Y60 in KCNE3. (**D**) Binding site of KCNE1 FTL and KCNE3 TVG. The putative H-bond between KCNQ1 Y267 and KCNE1 T58 is indicated by a dashed line. (**E**) Residue neighborhood around Y65 in KCNE1 and its homologous residue Y79 in KCNE3. (**F**) Occurrence of the Y267-T58 H-bond in MD simulations of the KCNQ1-KCNE1 RC and AO model.

The online version of this article includes the following source data and figure supplement(s) for figure 8:

**Source data 1.** Excel file with MD H-bond time series data used for *Figure 8* panel F.

**Figure supplement 1.** Double mutant cycle analysis for residue pair KCNQ1 Y267–KCNE1 T58.

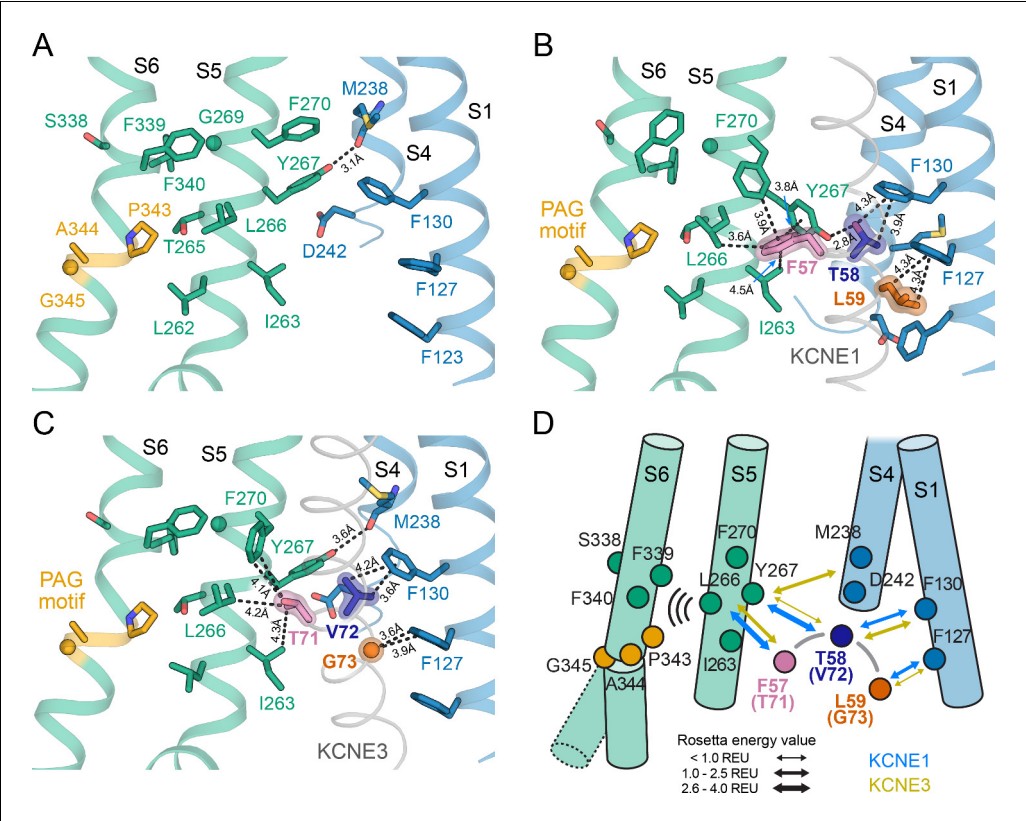

**Figure 9.** The activation motifs of KCNE1 and KCNE3 form distinct interactions with the VSD and PD that may induce different allosteric effects on S6. (**A**) KCNE1/3 binding cleft in unbound KCNQ1 (PDB: 6UZZ) (**Sun and MacKinnon, 2020**). Residues on S1, S4, and S5, which surround FTL in the KCNQ1-KCNE1 model and TVG in the KCNQ1-KCNE3 structure, as well as residues in the S6 helix are shown. (**B**) Predicted binding mode of the KCNE1 FTL as discussed in the text. Potentially interacting residues are indicated and their distances are labeled. (**C**) Binding mode of the KCNE3 TVG observed in the KCNQ1-KCNE3 structure (PDB: 6V00). The distances to potentially interacting residues are labeled. (**D**) Schematic representation of the interactions induced by binding of KCNE1 (blue arrows) and KCNE3 (olive arrows), respectively. The expected relative strength of an interaction computed with the Rosetta energy function is indicated by the arrow thickness (Rosetta energy unit, REU). KCNE1/3 binding may be allosterically coupled to S6 as supported by previously reported functional interactions of KCNE3 with S338, and KCNE1 with F339 and F340 (**Melman et al., 2004**; **Panaghie et al., 2006**). Those interactions may affect S6 kinking at the PAG motif and influence gate opening.

The online version of this article includes the following figure supplement(s) for figure 9:

**Figure supplement 1.** Rosetta energy calculations and dynamical network analysis suggest that KCNQ1 S5 makes distinct interactions with KCNE1 and is part of a putative allosteric network connecting KCNE1 with KCNQ1 S6.

**Figure supplement 2.** TMD sequence conservation within the KCNE family.

of KCNQ1 activation by KCNE1 is reduced by mutations Y267A and F270A, and to a smaller extent by L271A, I274A, and F275A (**Strutz-Seebohm et al., 2011**). Mutation F270A also altered the volume-dependency of $V_{1/2}$ changes resulting from substitution of KCNE1 T58 with amino acids of different size (**Strutz-Seebohm et al., 2011**). Furthermore, KCNE1 FTL was responsible for the suppression of constitutive currents in KCNQ1 channels with mutation I268A, even in cases of functionally decoupled voltage sensor and pore domains (**Barro-Soria et al., 2017**). These data are consistent with binding of FTL to this region on S5 leading to triggering of changes in the activation gate of KCNQ1. Indeed, dynamical network analysis of our MD simulations shows that S5 connects KCNE1 T58 with KCNQ1 S6 through several short pathways in a residue interaction network (**Figure 9—figure supplement 1A and B**). These allosteric interactions could alter the conformational dynamics of S6 around the PAG motif and influence gate opening.

There are several lines of experimental evidence that support this model. KCNQ1 residues F339 and F340, which are one helix turn before the PAG motif, were shown to be energetically coupled to KCNE1 T58 based on double mutant cycle analysis (*Strutz-Seebohm et al., 2011*; *Li et al., 2015*). This is in accord with our dynamical network analysis (*Figure 9—figure supplement 1A and B*) and with the observation that mutations at either F339, F340, or T58 produce similar functional outcomes (*Melman et al., 2004*). Furthermore, the phenotype of some mutations in S6 can be altered or rescued by KCNE1: non-functional KCNQ1 A341V mutant channels can be rendered functional when co-expressed with KCNE1 (*Mikuni et al., 2011*), and constitutively active F340W channels can be suppressed by mutation at KCNE1 T58 (*Panaghie et al., 2006*), which points to an effect of KCNE1 on the channel gate. Previously, these data were taken as evidence for a direct physical contact between T58 and S6, leading to a model in which KCNE1 lies close to or forms a part of the KCNQ1 PD (*Tapper and George, 2001*; *Melman et al., 2004*; *Strutz-Seebohm et al., 2011*; *Li et al., 2015*). However, our KCNQ1-KCNE1 models, in accord with structures of the related KCNQ1-KCNE3 complex (*Sun and MacKinnon, 2020*), indicate that KCNE1 is too far away to directly contact the middle region of S6. Instead, we propose an allosteric mechanism for control of KCNQ1 gating, in which binding of the FTL motif influences the conformational state of the S6 PAG motif by interacting with S5, which then directly impacts the key gating residues in S6 (*Figure 9D*, *Video 3*).

Comparison of the binding modes of KCNE1 and KCNE3 also suggests how the activation motifs of these proteins could trigger different effects on the KCNQ1 gate through mediation of S5. We observed different interactions are made by the KCNE1 FTL versus the KCNE3 TVG (*Figure 9B+C*, *Video 4*). KCNE1 F57 and L59 are much bulkier than the corresponding KCNE3 residues, T71 and G73, possibly leading to different steric effects on neighboring residues in KCNQ1 S1 and S5. Furthermore, the model predicts an H-bond between KCNE1 T58 and KCNQ1 Y267, which is absent for KCNE3 V72. Replacing V72 with a hydroxylated amino acid could restore this H-bond interaction, which offers an explanation why the KCNE3 V72T mutant leads to channel properties akin to KCNQ1-KCNE1 (*Melman et al., 2002*). Scoring with the Rosetta energy function suggests a larger binding free energy for FTL compared to TVG (*Figure 9D*, *Figure 9—figure supplement 1C*) and a decrease of binding by Ala mutations of residues in S5 (*Figure 9—figure supplement 1D*). Together these observations support the notion that FTL and TVG induce distinct interactions of varying strength with S5 that determine the effect of the KCNE subunits on the S6 gate.

The outlined mechanistic model agrees qualitatively with observations from *Barro-Soria et al., 2017*, who showed that KCNE1 and KCNE3 have different effects on the VSD and PD. While KCNE1 acts both on the VSD and PD, shifting S4 movement to more negative potentials and gate opening to more positive potentials, KCNE3 mainly affects the VSD (*Barro-Soria et al., 2017*), effectively eliminating gating by stabilizing the intermediate and fully activated VSD states (*Sun and MacKinnon, 2020*; *Kroncke et al., 2016*). The FTL and TVG motifs were found to determine whether the KCNE subunits affect the gate and the second S4 movement that is seen in voltage clamp fluorometry studies of KCNQ1-KCNE1 and correlates with the opening of KCNQ1-KCNE1 channels (*Barro-Soria et al., 2017*). This can be explained by the differences in the interaction modes between FTL and TVG seen in our models. Furthermore, significantly less contacts are observed for FTL or TVG with S4 (only KCNQ1 V241 is within van der Waals distance of either T58 or V72) (*Figure 8D*), which is consistent with the notion that these sites have minimal impact on S4 movement. This is in contrast to other residues in KCNE1 and KCNE3, one to two helix turns C-terminal to FTL and TVG, which make direct contacts with S4, explaining how both KCNE proteins affect S4 movement (*Panaghie and Abbott, 2007*; *Nakajo and Kubo, 2007*; *Rocheleau and Kobertz, 2008*;

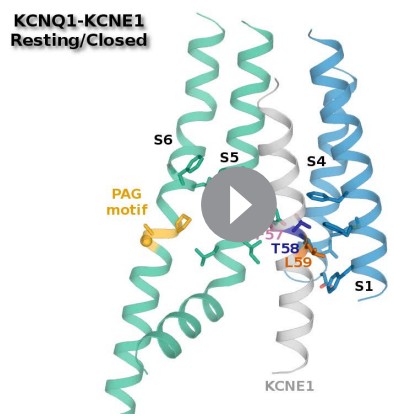

**Video 4.** Animation and comparison of the interaction sites of the activation motifs from KCNE1 and KCNE3. https://elifesciences.org/articles/57680#video4

*Osteen et al., 2010*; *Ruscic et al., 2013*; *Barro-Soria et al., 2014*; *Barro-Soria et al., 2015*).

Despite different interactions of KCNQ1 with the activation motifs from KCNE1 and KCNE3, we observed impressive similarity in their global TMD-binding modes. This observation led us to wonder if this feature is conserved for other KCNEs. Some support for this hypothesis comes from the overall high degree of sequence homology of the TMD within the KCNE family (11–60% identity, 45–78% similarity) (*Figure 9—figure supplement 2*). It is noteworthy that Y46 in KCNE1, which we identified as a high impact mutation site, is conserved in all KCNE proteins. Perhaps, this site is part of a channel docking/anchoring mechanism shared by all KCNE family members. A region with considerably less sequence homology exists in the middle of the TMD and coincides with the FTL motif from KCNE1. We speculate that different KCNEs might have evolved different amino acid sequences for this motif in order to elicit different functional responses in KCNQ1 and other ion channels. This sequence motif may therefore represent a common regulatory element in KCNEs.

There is relatively little structure-function data available for other KCNEs to inform on the degree of conservation of their TMD-binding modes. Using cysteine scanning mutagenesis and analyzing the pattern of functional perturbations in KCNE2, *Wang et al., 2012* developed a schematic model for the orientation of the KCNE2 TMD within the KCNQ1 channel complex. KCNQ1 C331 was observed to form disulfide bonds to different positions in KCNE1 (F54) and KCNE2 (M59, corresponding to F53 in KCNE1), which trapped the channel in open and closed states, respectively. While this result could imply a rotation of the KCNE subunit during channel open-to-close transition, it could also indicate a different TMD orientation. In contrast to this hypothesis, our KCNQ1-KCNE1 models and the KCNQ1-KCNE3 structures fail to reveal a significant rotation for the KCNE subunit between open and closed channel conformations. Furthermore, in these structures, F53 and F54 (L67 and F68 in KCNE3) have approximately the same distance to C331 (KCNE1: F53 – 10.1 Å, F54 – 13.2 Å; KCNE3: L67 – 9.4 Å, F68 – 12.5 Å). We suggest that the proposed KCNE1 binding pose of this work could potentially satisfy both disulfide bond restraints, and that experimental KCNE-specific differences in disulfide bond formation could be the result of local structural differences in and around the activation triplet between KCNE1 and KCNE2. Another study by *Li et al., 2015* identified functional interactions between the KCNQ1 PD (F275, F340) and the middle residue in the KCNE2 activation triplet (I64). This is consistent with our proposed allosteric mechanism between those corresponding sites in KCNE1 and suggests a similar mechanism may exist in KCNE2.

In summary, our results imply that all KCNE subunits interact with KCNQ1 by adopting a common binding mode within their TMD. Moreover, our experimental/computational models of the KCNQ1-KCNE1 channel complex provide an explanation for how the KCNE1 TMD interacts with the KCNQ1 channel and controls KCNQ1 activation through its intramembrane FTL motif. Binding of the FTL motif appears to affect the channel gate through allosteric interactions with S6 that are mediated by S5. Differences in their interactions with S5 appear to explain the different effects of KCNE1 and KCNE3 on the gate in KCNQ1.

## Materials and methods

**Key resources table**

| Reagent type (species) or resource | Designation | Source or reference | Identifiers | Additional information |
|---|---|---|---|---|
| Cell line (*Cricetulus griseus*) | CHO-K1 | ATCC Manassas, VA | RRID:SCR_001672 | Expression cell line |
| Gene (*Homo sapiens*) | KCNQ1 | HUGO Gene Nomenclature Committee (HGNC) | Gene ID: 3784; HGNC:629 | |
| Gene (*Homo sapiens*) | KCNE1 | HUGO Gene Nomenclature Committee (HGNC) | Gene ID: 3753; HGNC:624 | |
| Commercial assay, kit | Nucleobond Xtra Maxi EF | Macherey-Nagel Inc, Bethlehem, PA | Cat. # NC00089196 | Used to isolate DNA |
| Recombinant DNA reagent | pIRES2-EGFP | BD Biosciences-Clontech Mountain View, CA | | Used to express KCNQ1 |

*Continued on next page*

*Continued*

| Reagent type (species) or resource | Designation | Source or reference | Identifiers | Additional information |
|---|---|---|---|---|
| Recombinant DNA reagent | pIRES2-Scarlet | PMID:19687231 | | Used to express KCNE1 |
| Commercial assay, kit | QuikChange II XL | Agilent technologies Santa Clara, CA | Cat. # 200521 | Used to generate channel protein variants |
| Chemical compound, drug | Fugene six transfection reagent | Promega Corporation Madison, WI | Cat. # E2691 | Used to transfect cDNAs |
| Chemical compound, drug | JNJ 303 | TOCRIS Minneapolos, MN | Cat. # 3899 | Chemical compound, drug |
| Software, algorithm | Excel | Microsoft Redmon, WA | | Data analysis |
| Software, algorithm | PatchController | Nanion Technologies Munich, Gemany | | Electrophysiology data collection |
| Software, algorithm | DataController | Nanion Technologies Munich, Gemany | | Electrophysiology data analysis |
| Software, algorithm | Clampex | Axon Instruments Sunnyvale, CA | RRID:SCR_011323 | Electrophysiology data collection |
| Software, algorithm | Clampfit | Axon Instruments Sunnyvale, CA | RRID:SCR_011323 | Electrophysiology data analysis |
| Software, algorithm | Sigmaplot | SPSS San Jose, CA | RRID:SCR_003210 | Data plotting |
| Software, algorithm | GraphPad Prism | GraphPad Software San Diego, CA | RRID:SCR_000306 | Data analysis and plotting |
| Software, algorithm | ROSETTA (version 3.9) | PMID:21187238 URL: https://www.rosettacommons.org/ | RRID:SCR_015701 | Protein-protein docking |
| Software, algorithm | MolProbity | PMID:17452350 URL: http://molprobity.biochem.duke.edu | RRID:SCR_014226 | Analysis of docking models |
| Software, algorithm | CHARMM-GUI | PMID:25130509 URL: http://www.charmm-gui.org | | Preparation of MD system |
| Software, algorithm | AMBER 16 | PMID:16200636 URL: https://ambermd.org | RRID:SCR_014230 | Program for execution of MD simulations |
| Software, algorithm | CPPTRAJ | PMID:26583988 URL: https://ambermd.org/AmberTools.php | | Tools for analysis of MD trajectories |
| Software, algorithm | Antechamber | PMID:16458552 URL: https://ambermd.org/AmberTools.php | | Parameterization of PIP2 lipid molecule |
| Software, algorithm | Gaussian 09 | Gaussian, Inc, Wallingford CT URL: https://gaussian.com | RRID:SCR_014897 | Parameterization of PIP2 lipid molecule |
| Software, algorithm | PyMOL | The PyMOL Molecular Graphics System, Version 2.0 Schrödinger, LLC URL: https://pymol.org/ | RRID:SCR_000305 | Visualization of KCNQ1-KCNE1 models |
| Software, algorithm | VMD | PMID:8744570 URL: https://www.ks.uiuc.edu/Research/vmd/ | RRID:SCR_001820 | Visualization of MD simulations |
| Software, algorithm | HOLE | PMID:9195488 URL: http://www.holeprogram.org | | Calculation of channel pore radius |
| Software, algorithm | NACCESS | URL: http://wolf.bms.umist.ac.uk/naccess/ | | SASA calculation |

*Continued on next page*

*Continued*

| Reagent type (species) or resource | Designation | Source or reference | Identifiers | Additional information |
|---|---|---|---|---|
| Software, algorithm | NetworkView Plugin for VMD | PMID:22982572 URL: https://www.ks.uiuc.edu/ Research/ vmd/ plugins/networkview/ | | Network analysis of MD simulations |
| Software, algorithm | Anaconda | Anaconda Software Distribution. Computer software. Vers. 2–2.4.0. Anaconda, Inc URL: https://www.anaconda.com | | Data plotting |

## Computational docking of KCNE1 to KCNQ1

Molecular models of the KCNQ1-KCNE1 complex were developed by computational docking with Rosetta (version 3.9) (*Leaver-Fay et al., 2011*; *Figure 3—figure supplement 2*). KCNE1 was docked to human KCNQ1 models of the closed state with the VSD and PD in resting and closed (RC) conformations and to models of the open state with the VSD and PD in fully activated and open (AO) conformations. Molecular modeling was guided by contact restraints derived from the results of disulfide crosslinking and site-directed mutagenesis experiments, which were obtained in this study or collected from previous literature reports (*Tapper and George, 2001*; *Chung et al., 2009*; *Wang et al., 2011*; *Chan et al., 2012*; *Wang et al., 2012*; *Strutz-Seebohm et al., 2011*; *Li et al., 2015*). Two separate restraint lists for docking of KCNE1 to either the closed or open KCNQ1 model were compiled based on the functional annotation of the channel state under the experimental conditions. A crosslink restraint was assigned the closed or open conformation if it trapped the channel in the closed or open state, respectively, or both conformations if the channel state was unclear. For example, Chung and coworkers (*Chung et al., 2009*) reported that disulfide crosslinks KCNE1 K41–KCNQ1 I145 and KCNE1 L42–KCNQ1 V324 favored the open state, whereas crosslinks KCNE1 K41–KCNQ1 V324 and KCNE1 L42–KCNQ1 I145 stabilized the closed state. Similarly, *Wang et al., 2011* concluded, in the closed state, KCNQ1 residues T144, I145, Q147 are preferably crosslinked to KCNE1 R36-E43, whereas in the open state, those positions in KCNQ1 crosslink with KCNE1 G40, and KCNQ1 Q147 can be disulfide-bonded to KCNE1 R36, G38, and K41. Complete restraint lists for development of the closed and open KCNQ1-KCNE1 model can be found in *Supplementary file 1 – Table 2 and 3*, respectively.

Prior to docking, the TMD and a short stretch of the N-terminal juxtamembrane domain of the ten deposited models of the KCNE1 NMR structure (PDB: 2K21) (*Kang et al., 2008*), hereupon termed KCNE1 TMD (residues S37-L71), were energy-minimized with Rosetta using the Rosetta-Membrane (*Yarov-Yarovoy et al., 2006*; *Barth et al., 2007*) energy function. KCNE1 TMD was then placed near the transmembrane region of our previously published KCNQ1 homology models of the RC and AO state (*Kuenze et al., 2019*), and a total of 40,000 KCNQ1-KCNE1 models were generated for each state using the Rosetta protein-protein docking algorithm (*Gray et al., 2003*; *Gray, 2006*). The disulfide crosslinks and additional contact information were implemented as Cα-atom pair distance restraints with an upper bound of 12 Å, which corresponds to the length of an extended disulfide crosslink (7 Å) plus an additional 5 Å padding to account for the effect of protein flexibility. For Cd(II)-cysteine crosslinks (*Tapper and George, 2001*), the upper bound distance was increased to 15 Å (corresponding to 10 Å theoretical distance plus 5 Å padding). Distance restraints were evaluated with a harmonic penalty function that was zero below 12 Å (15 Å for Cd(II)-bridged crosslinks) and grew quadratically beyond that distance. The KCNQ1-KCNE1 docking models were filtered by a combination of score criteria (interface score <0 REU, $\Delta G_{Binding}$ <0 REU, atom pair restraint score <350 REU) and the remaining models were sorted by the binding energy between KCNQ1 and KCNE1 ($\Delta G_{Binding}$). The 1000 best-scoring KCNQ1-KCNE1 models were then used as input structures for a subsequent round of docking to generate additional 40,000 model complexes. This alternating docking-filtering procedure (*Figure 3—figure supplement 2*) was iterated until the change in $\Delta G_{Binding}$ averaged over the ten lowest-energy models from iteration to iteration converged to less than 5%. In subsequent iterations, the atom pair restraint score cutoff was gradually decreased from 350 REU to 100 REU in the last docking round in order to apply a more restrictive experimental filter. In addition, the allowed range of translational and rotational perturbations at the

beginning of each docking run was decreased from initially 3 Å / 8° to 1 Å / 3° to enable a more fine-grained conformational sampling. Docking calculations converged after five to seven iterations, and representative models were chosen by RMSD-based clustering of the 5000 best-scoring models, followed by visual inspection and MolProbity (*Davis et al., 2007*) analysis of the 10–20 lowest-energy models from the ten largest clusters. The model with the best combined $\Delta G_{Binding}$ and MolProbity scores from the largest model cluster as well as two additional low-energy models from the same cluster were selected for further MD analysis.

For MD simulations, KCNQ1-KCNE1 models were prepared with a stoichiometry of 4:2 KCNQ1: KCNE1 subunits. This appears to represent the predominant stoichiometry on the surface of mammalian cells (*Plant et al., 2014*), although the possibility of multiple stoichiometries ranging from 4:1 to 4:4 has been discussed (*Plant et al., 2014*; *Morin and Kobertz, 2008*; *Nakajo et al., 2010*; *Murray et al., 2016*; *Chen et al., 2003b*). The docked conformation of KCNE1 was duplicated and aligned with the opposite half of KCNQ1 by 180° rotation around the central channel axis. Subsequently, the interface of both KCNE1 molecules with KCNQ1 was relaxed by sidechain rotamer repacking and energy minimization of all backbone and sidechain degrees of freedom in Rosetta. Representative KCNQ1-KCNE1 complex models (with 4:2 stoichiometry) are provided with the supporting material to this paper (*Supplementary file 2* and *3*) and can be obtained from PDB-Dev under accession number PDBDEV_00000042. Restraints used in docking and starting model coordinates can be obtained from https://doi.org/10.5281/zenodo.3598943.

## MD simulations of KCNQ1-KCNE1 models

MD simulations of KCNQ1-KCNE1 models were performed in explicit phospholipid membranes at 310 K with AMBER16 (*Case et al., 2016*) employing the ff14SB (*Maier et al., 2015*) force field for proteins and the Lipid17 force field (*Gould IR, Skjevik AA, Dickson CJ, Madej BD, Walker RC, 2018, "Lipid17: A Comprehensive AMBER Force Field for the Simulation of Zwitterionic and Anionic Lipids", manuscript in preparation*). As starting conformations for MD, the Rosetta model with the best combined $\Delta G_{Binding}$ and MolProbity scores and two additional low-energy models from the ensemble of Rosetta docking models were selected, and prepared with a 4:2 KCNQ1:KCNE1 stoichiometry, as described above. Models were aligned to the membrane normal using the PPM webserver (*Lomize et al., 2012*) and embedded into bilayers of POPC (palmitoyloleoyl-phosphatidylcholine) and PIP2 (phosphatidyl-4,5-bisphosphate) (~280 lipids per leaflet) using the membrane builder tool of the CHARMM-GUI website (*Wu et al., 2014*). A TIP3P water layer with 24 Å thickness containing 150 mM of KCl was added on either side of the membrane. In addition, four $K^+$ ions were placed in the channel selectivity filter at positions inferred from the X-ray structure of $K_V$1.2–2.1 (PDB: 2R9R). Bilayers contained 10 mol% of PIP2 in the inner leaflet which comprised equal numbers of $C4-PO_4$- and $C5-PO_4$-mono-protonated PIP2 molecules with stearoyl and arachidonoyl conjugations at the sn-1 and sn-2 position. The geometry of PIP2 was optimized with Gaussian 09 (Gaussian, Inc, Wallingford CT) on the B3LYP/6–31G** level of theory, and assignment of AMBER atom types and calculation of RESP charges was done with Antechamber (*Wang et al., 2006*). Bond and angle parameters of the protonated $C4-PO_4$ or $C5-PO_4$ group in PIP2 were adjusted to values previously reported for phosphorylated amino acids (*Homeyer et al., 2006*) to avoid simulation instabilities. SHAKE (*Ryckaert et al., 1977*) bond length constraints were applied to all bonds involving hydrogen. Nonbonded interactions were evaluated with a 10 Å cutoff, and electrostatic interactions were calculated by the particle-mesh Ewald method (*Darden et al., 1993*).

Each MD system was first minimized for 15,000 steps using steepest descent followed by 15,000 steps of conjugate gradient minimization. With protein and ions restrained to their initial coordinates, the lipid and water were heated to 50 K over 1000 steps with a step size of 1 fs in the NVT ensemble using Langevin dynamics with a rapid collision frequency of 10,000 $ps^{-1}$. The system was then heated to 100 K over 50,000 steps with a collision frequency of 1000 $ps^{-1}$ and finally to 310 K over 200,000 steps and a collision frequency of 100 $ps^{-1}$. After changing to the NPT ensemble, restraints on ions were gradually removed over 500 ps and the system was equilibrated for another 5 ns at 310 K with weak positional restraints (with a force constant of 1 kcal $mol^{-1}$ $Å^{-2}$) applied to protein Cα atoms. The protein restraints were then gradually removed over 10 ns, and production MD was conducted for 450 ns using a step size of 2 fs, constant pressure periodic boundary conditions, anisotropic pressure scaling and Langevin dynamics. Four independent simulations were carried out for the RC and AO KCNQ1-KCNE1 model yielding 1.80 μs of total MD data for each state.

Representative snapshots from the MD simulations can be obtained from https://doi.org/10.5281/zenodo.3598943.

## Analysis of KCNQ1-KCNE1 MD simulations

Analysis of MD trajectories with CPPTRAJ (version 18.0) (*Roe and Cheatham, 2013*) included calculation of Cα-atom root-mean-square deviations (Cα-RMSD), enumeration of protein-protein hydrogen bonds, measurement of residue pair distances in the VSD, and counting of intermolecular contacts between KCNQ1 and KCNE1. Residue contact numbers were calculated by counting within a 4 Å radius of a given KCNQ1 or KCNE1 residue the number of heteroatoms from the other protein binding partner (i.e. the contact number of a KCNE1 residue was calculated by counting the number of atoms from KCNQ1 that were within 4 Å and vice versa). The residue contact number was then averaged over both KCNE1 molecules in all MD trajectories of the RC or AO KCNQ1-KCNE1 model, respectively.

Measurement of the channel pore radius was carried out with the HOLE program (*Smart et al., 1996*) using snapshots of KCNQ1 taken at one ns intervals during the last 400 ns of MD. Calculation of the solvent-accessible surface area (SASA) of KCNE1 was conducted using NACCESS (*Hubbard and Thornton, 1993*). In addition, computation of the binding free energy ($\Delta G_{Binding}$) between KCNQ1 and KCNE1 was carried out using the *MMPBSA.py* program (*Miller et al., 2012*). A total of 2660 KCNQ1-KCNE1 conformations sampled at 150 ps intervals from the last 400 ns of a MD trajectory were processed to compute the molecular mechanics potential energies and solvation free energies in the MMPBSA procedure (*Kollman et al., 2000*). The solvation free energy contribution to $\Delta G_{Binding}$ was calculated using a continuum Poisson-Boltzmann (PB) model for channel proteins as described in *Xiao et al., 2017*. The entropic contribution to $\Delta G_{Binding}$ was estimated by applying the quasi-harmonic approximation (QHA) (*Karplus and Kushick, 1981*), and 26,600 KCNQ1-KCNE1 conformations were used for this analysis. $\Delta G_{Binding}$ of KCNQ1-KCNE1 was then calculated as the difference between the free energy of the KCNQ1-KCNE1 complex and the sum of the KCNQ1 and KCNE1 free energies.

Dynamical network analysis was performed with the Network View plugin (*Eargle and Luthey-Schulten, 2012*) in VMD (*Humphrey et al., 1996*). A node in the network was assigned to every amino acid in KCNQ1 and KCNE1 centered at their Cα atom. Network edges were defined between nodes whose residues were within 4.5 Å distance for at least 75% of the MD trajectory. The last 400 ns of simulation were used for the analysis. Edge weights were derived from the pairwise residue correlation matrix calculated with the program Carma (*Glykos, 2006*). Network communities were determined with the Girvan-Newman algorithm (*Girvan and Newman, 2002*) implemented in the program gncommunities as part of the Network View plugin (*Eargle and Luthey-Schulten, 2012*).

## Control docking calculations for KCNE1 and KCNE3

To check whether use of homology models of KCNQ1 as input for docking affected structure prediction of the KCNQ1-KCNE1 complex, docking calculations were also carried out with the cryo-EM-determined structure of human KCNQ1 (*Sun and MacKinnon, 2020*), which was released after the KCNQ1-KCNE1 models of this work were completed. In addition, docking calculations were performed with KCNE3 to assure that our computational protocol could recapitulate the experimentally observed structure for the KCNQ1-KCNE3 complex.

Prior to docking, the cryo-EM-determined open state structure of human KCNQ1 bound to calmodulin and KCNE3 (PDB: 6V01) (*Sun and MacKinnon, 2020*) was minimized with Rosetta using the FastRelax protocol (*Conway et al., 2014*) guided by the cryo-EM density map (*Wang et al., 2016*). C4-symmetry was enforced with the help of a symmetry definition file (*DiMaio et al., 2011*), and positional restraints on the protein backbone atoms were gradually ramped down during five repeats of FastRelax. Subsequently, 10 models of the NMR-determined KCNE1 TMD structure (S37-L71) (PDB: 2K21) (*Kang et al., 2008*) were docked to the cryo-EM AO model as described above using experimental restraints for the open channel state. The KCNQ1-KCNE1 model with the best $\Delta G_{Binding}$ score was deemed the final model and compared to the model developed in this work by docking KCNE1 to the Rosetta homology model of KCNQ1. Model similarity was evaluated by calculating the heavy-atom RMSD for the residues in the KCNQ1-KCNE1 interface within 10 Å of any residue on the other protein, and by computing the fraction of recovered contacts. A contact was

defined as every residue in KCNQ1 (KCNE1) that was within 5 Å of a residue in KCNE1 (KCNQ1) in the reference model.

Similarly, ten models of the KCNE3 TMD NMR structure (P51-V85) (PDB: 2NDJ) (*Kroncke et al., 2016*) were docked to both the cryo-EM and Rosetta KCNQ1 AO model using the same computational protocol and published experimental restraints for the KCNQ1-KCNE3 complex (*Kroncke et al., 2016*). Similarity between the docking model and experimental KCNQ1-KCNE3 structure was assessed based on the all-atom interface RMSD and fraction of native contacts recovered.

## Mammalian cell culture

Chinese hamster ovary cells (CHO-K1, CRL 9618, American Type Culture Collection, Manassas VA, USA) were grown in F-12 nutrient medium (GIBCO/Invitrogen, San Diego, CA, USA) supplemented with 10% fetal bovine serum (ATLANTA Biologicals, Norcross, GA, USA), penicillin (50 units/mL), streptomycin (50 µg/mL) at 37°C in 5% $CO_2$. The identity of CHO-K1 cells was certified by American Type Culture Collection using Cytochrome C Oxidase (COI) assay testing. Cells were negative for mycoplasma contamination and are regularly tested using the MycoAlert PLUS Mycoplasma Detection Kit (Lonza, Rockville, MD, USA). Unless stated otherwise, all tissue culture media was obtained from Life Technologies, Inc (Grand Island, NY, USA). CHO-K1 cells constitutively expressing human KCNE1 (designated CHO-KCNE1 cells) were generated using the FLP-in system (Thermo Fisher Scientific, Waltham, MA, USA) and maintained under selection with hygromycin B (600 µg/mL) as described previously (*Vanoye et al., 2018*).

## Plasmids and heterologous expression

KCNQ1 cDNA (GenBank accession AF000571) was engineered in the pIRES2-EGFP expression vector (BD Biosciences-Clontech, Mountain View, CA, USA) or a modified pIRES2-mScarlet vector, and KCNE1 cDNA (GenBank accession L28168) was cloned into a pIRES2-DsRed-MST vector as described previously (*Vanoye et al., 2018*; *Lundquist et al., 2005*; *Manderfield and George, 2008*). These vectors allowed co-expression of KCNQ1 and KCNE1 with fluorescent proteins as means for tracking successful cell transfection. Mutants of KCNQ1 and KCNE1 were generated using the QuikChange II XL system (Agilent technologies, Santa Clara, CA, USA). Correctness of the KCNQ1 and KCNE1 coding region was checked by DNA sequencing (Eurofins Genomics, Louisville, KY, USA), and plasmid DNA was amplified using an endotoxin-free plasmid preparation method (Nucleobond Xtra Maxi EF, Macherey-Nagel Inc, Bethlehem, PA, USA). Transfection of plasmid DNA encoding KCNQ1 and KCNE1 WT or variants into CHO-K1 cells for manual patch clamp experiments was performed using Fugene as previously described (*Vanoye et al., 2009*). Transfection of KCNQ1 WT and variant cDNA into CHO-KCNE1 cells (i.e. CHO-K1 cells with stable expression of KCNE1) and the transfection of KCNQ1 WT and KCNE1 WT or mutant cDNA into CHO-K1 cells for automated patch clamp recordings were done by electroporation using the Maxcyte STX system (MaxCyte Inc, Gaithersburg, MD, USA) as described previously (*Vanoye et al., 2018*).

## Electrophysiology

Automated patch clamp experiments were performed using the Syncropatch 768 PE platform (Nanion Technologies, Munich, Germany) equipped with single-hole, 384-well recording chips with medium resistance (2–4 MΩ). Pulse generation and data collection were carried out with PatchController384 V.1.3.0 and DataController384 V1.2.1 software (Nanion Technologies, Munich, Germany). Whole-cell currents were filtered at 3 kHz and acquired at 10 kHz. The access resistance and apparent membrane capacitance were estimated using built-in protocols. Whole-cell currents were recorded at room temperature in the whole-cell configuration from −80 to +60 mV (in 10 mV steps) at 1990 ms after the start of the voltage pulse from a holding potential of −80 mV. The external bath solution contained: 140 mM NaCl, 4 mM KCl, 2 mM $CaCl_2$, 1 mM $MgCl_2$, 10 mM HEPES, 5 mM glucose, pH 7.4. The internal solution contained: 60 mM KF, 50 mM KCl, 10 mM NaCl, 10 mM HEPES, 10 mM EGTA, 2 mM ATP-$K_2$, pH 7.2. Whole-cell currents were not leak-subtracted. The contribution of background currents was determined by recording before and after addition of 20 µM of the $I_{Ks}$ blocker HMR1556. Recordings with measurable outward current were examined to verify block by HMR1556. Only HMR1556-sensitive currents and recordings meeting the following criteria

were used in data analysis: seal resistance $\geq 0.5$ G$\Omega$, series resistance $\leq 20$ M$\Omega$, capacitance $\geq 1$ pF, voltage-clamp stability (defined as the standard error for the baseline current measured at the holding potential for all test pulses being <10% of the mean baseline current). Current-voltage (I-V) relationships were derived for all cell recordings meeting these criteria.

Oxidation-state dependent electrophysiological recordings of cysteine mutants of KCNQ1 (V141C, I274C) and KCNE1 (L45C, V47C, L48C) as well as the corresponding control experiments were done by manual patch clamp measurements. Whole-cell currents were recorded at room temperature (20–23˚C) using Axopatch 200 and 200B amplifiers (Molecular Devices Corp., Sunnyvale, CA, USA) in the whole-cell configuration of the patch clamp technique (*Hamill et al., 1981*). Pulse generation was performed with Clampex 10.0 (Molecular Devices Corp., Sunnyvale, CA, USA). Whole-cell currents were filtered at 1 kHz and acquired at 5 kHz. The access resistance and apparent membrane resistance were estimated using an established protocol (*Lindau and Neher, 1988*). Whole-cell currents were not leak-subtracted. Whole-cell currents were measured from −80 to +60 mV (in 10 mV steps) at 1990 ms after the start of the voltage pulse from a holding potential of −80 mV. The external bath solution contained: 132 mM NaCl, 4.8 mM KCl, 1.2 mM MgCl$_2$, 1 mM CaCl$_2$, 5 mM glucose, pH 7.4. The internal solution contained: 110 mM K$^+$-aspartate, 1 mM CaCl$_2$, 10 mM HEPES, 11 mM EGTA, 1 mM MgCl$_2$, 2 mM ATP-K$_2$, pH 7.3. The pipette solution was diluted 5–10% to prevent induction of swelling-activated currents. Patch pipettes were pulled from thick-wall borosilicate glass (World Precision Instruments, Inc, Sarasota, FL, USA) with a multistage P-97 Flaming-Brown micropipette puller (Sutter Instruments Co., San Rafael, CA, USA) and heat-polished with a Micro Forge MF 830 (Narashige, Japan). After heat polishing, the resistance of the patch pipettes was 3–5 M$\Omega$ in the control recording solution. As a reference electrode, a 2% agar-bridge with a composition similar to the control bath solution was utilized. Junction potentials were zeroed with the filled pipette in the bath solution. Unless otherwise stated, all chemicals were obtained from Sigma-Aldrich (St. Louis, MO, USA).

The possibility for disulfide bond formation between KCNQ1 and KCNE1 cysteine mutants was tested by perfusing cells with 10 mM 1,4-dithiothreitol (DTT, reducing) or 100–350 µM Cu(II)-phenanthroline (Cu-phen, oxidizing) in the external bath solution and measuring whole-cell currents as described above. Because the I$_{Ks}$ current when analyzed by whole-cell voltage clamp in mammalian cells exhibits rundown (see Supplemental Figure 2 in *Vanoye et al., 2018*), which varies in both rate and magnitude, thus confounding experiments on a single cell in which there is a time lapse between conditions and treatments, we recorded from multiple different cells for each treatment (Control, +DTT, +Cu-phen) to average the cell-to-cell variability. In addition, cells were recorded for all treatments from a specific cell transfection batch and from at least three distinct transfections for each KCNQ1-KCNE1 combination.

## Electrophysiological data analysis

Data were collected for each experimental condition from at least three transfections and analyzed and plotted using DataController384 V1.2.1 (Nanion Technologies, Munich, Germany), Clampfit V10.4 (Molecular Devices Corp.), Excel (Microsoft Office 2013, Microsoft), SigmaPlot 2000 (Systat Software, Inc, San Jose, CA, USA) and OriginPro 2016 (OriginLab, Northampton, MA, USA) software. Whole-cell currents were normalized for membrane capacitance and results expressed as mean ± SEM. The number of cells used for each experimental condition and the threshold for statistical significance (p<0.001) are given in the figure legends or table footnotes. Additional custom semi-automated data handling routines were used for rapid analysis of current density, voltage-dependence of activation, and gating kinetics. The voltage-dependence of activation was determined only for cells with mean current density greater than the background current amplitude. Since normalized KCNQ1-KCNE1 tail currents do not saturate at the potentials tested nor would they saturate at more depolarized potentials (see Figure 3C in *Wang et al., 2020* and Figure 3C in *Wang et al., 2012*), we refer to the V$_{1/2}$ value determined by curve fitting as the 'apparent' activation V$_{1/2}$ (V$_{1/2app}$).

## Acknowledgements

This work was supported by NIH grants R01 HL122010 and R01 GM080403. GK was supported by fellowships from the German Research Foundation (KU 3510/1–1) and the American Heart

Association (18POST34080422). EFM was supported by NIH training grant T32 GM065086. KRB was supported by NIH training grant T32 GM008320. This work was conducted using the resources of the Advanced Computing Center for Research and Education (ACCRE) at Vanderbilt University. JM further acknowledges the Deutsche Forschungsgemeinschaft (DFG, German Research Foundation) through SFB1423, project number 421152132.

## Additional information

### Funding

| Funder | Grant reference number | Author |
|---|---|---|
| National Institutes of Health | R01 HL122010 | Charles R Sanders<br>Alfred L George Jr<br>Jens Meiler |
| National Institutes of Health | R01 GM080403 | Jens Meiler |
| American Heart Association | 18POST34080422 | Georg Kuenze |
| Deutsche Forschungsgemeinschaft | KU 3510/1-1 | Georg Kuenze |
| Deutsche Forschungsgemeinschaft | SFB1423/421152132 | Jens Meiler |
| National Institutes of Health | T32 GM065086 | Eli F McDonald |
| National Institutes of Health | T32 GM008320 | Kathryn R Brewer |

The funders had no role in study design, data collection and interpretation, or the decision to submit the work for publication.

### Author contributions

Georg Kuenze, Conceptualization, Data curation, Formal analysis, Validation, Investigation, Visualization, Methodology, Writing - original draft, Writing - review and editing; Carlos G Vanoye, Conceptualization, Data curation, Formal analysis, Validation, Investigation, Visualization, Methodology, Writing - review and editing; Reshma R Desai, Sneha Adusumilli, Kathryn R Brewer, Hope Woods, Eli F McDonald, Data curation, Investigation, Methodology, Writing - review and editing; Charles R Sanders, Conceptualization, Resources, Data curation, Supervision, Funding acquisition, Validation, Project administration, Writing - review and editing; Alfred L George Jr, Jens Meiler, Conceptualization, Resources, Supervision, Funding acquisition, Validation, Project administration, Writing - review and editing

### Author ORCIDs

Georg Kuenze (ID) https://orcid.org/0000-0003-1799-346X
Eli F McDonald (ID) https://orcid.org/0000-0002-0572-330X
Charles R Sanders (ID) https://orcid.org/0000-0003-2046-2862

### Decision letter and Author response

Decision letter https://doi.org/10.7554/eLife.57680.sa1
Author response https://doi.org/10.7554/eLife.57680.sa2

## Additional files

### Supplementary files

• Supplementary file 1. Table 1: Parameters from Boltzmann function fits of normalized activation curves of KCNQ1-KCNE1 WT and cysteine mutants which were used in crosslinking experiments. Table 2: Distance restraints used in generation of the closed state KCNQ1-KCNE1 docking model. Table 3: Distance restraints used in generation of the open state KCNQ1-KCNE1 docking model. Table 4: MolProbity statistics for KCNQ1-KCNE1 Rosetta models. Table 5: KCNQ1-KCNE1 residue

contacts observed in MD simulations of the KCNQ1-KCNE1 RC and AO channel models. Table 6: Biophysical properties of $I_{Ks}$ channels formed with KCNQ1 or KCNE1 mutants.

- Supplementary file 2. PDB coordinates of the KCNQ1-KCNE1 RC (closed state) docking model.
- Supplementary file 3. PDB coordinates of the KCNQ1-KCNE1 AO (open state) docking model.
- Transparent reporting form

### Data availability

The structural models developed in this work have been deposited in PDB-Dev under accession code PDBDEV_00000042 and are included as supplementary files 2 and 3 to this manuscript. The experimental restraints and starting model coordinates used for docking, as well as representative snapshots from the MD simulations have been deposited under: https://doi.org/10.5281/zenodo.3598943. All electrophysiology data generated and analyzed during this study are included in the manuscript or the supplementary source data files. All data needed to evaluate the conclusions in the paper are present in the paper and/or in the figure supplements and supplementary material.

The following dataset was generated:

| Author(s) | Year | Dataset title | Dataset URL | Database and Identifier |
| --- | --- | --- | --- | --- |
| Kuenze G, Vanoye CG, Desai RR, Adusumilli S, Brewer KR, Woods H, McDonald EF, Sanders CR, George AL, Meiler J | 2020 | Allosteric Mechanism for KCNE1 Modulation of KCNQ1 Potassium Channel Activation | https://doi.org/10.5281/zenodo.3598943 | Zenodo, 10.5281/zenodo.3598943 |

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
