## [Decision Letter]

**Acceptance summary:**

This paper presents an experimentally validated model of the IKs channel, KCNQ1+E1 in two states, Activated/Open and Resting /Closed, based in large part on an extensive review of experimentally-derived constraints. The models reported, deposited in PDB-dev will be of great value to the community. By comparing to the cryo-EM structure of KCNQ1+E3, the structural origin of profound functional differences with important physiological consequences is revealed.

**Decision letter after peer review:**

Thank you for submitting your article "Allosteric Mechanism for KCNE1 Modulation of KCNQ1 Potassium Channel Activation" for consideration by *eLife*. Your article has been reviewed by three peer reviewers, including Lucie Delemotte as the Reviewing Editor and Reviewer #1, and the evaluation has been overseen by Kenton Swartz as the Senior Editor. The following individual involved in review of your submission has agreed to reveal their identity: Leon D Islas (Reviewer #3).

The reviewers have discussed the reviews with one another and the Reviewing Editor has drafted this decision to help you prepare a revised submission.

KCNQ1 channels are a principal component of the slow-activating repolarization current in the heart. The slow activation character is produced by an interaction with the KCNE1 regulatory subunit. Although a well-characterized interaction, the molecular details of this are still controversial. The paper by Kuenze et al. investigates the molecular details of the association between the KCNQ1 channel and its regulatory subunit, KCNE1. Through molecular modeling and protein-protein docking, the authors arrive at a proposed mode of interaction between KCNE1 and models of the closed and open KCNQ1 channel. These detailed molecular interactions then are validated via a varied number of experiments, including mutagenesis, electrophysiological analysis of the effects of mutations and cysteine-cysteine crosslinking. Contrary to previous belief, the FTL motif in KCNE1 is not directly interacting with S6. Instead, the authors propose that FTL affects S6 via a allosteric network via S5. Comparing their KCNQ1/KCNE1 models with the KCNQ1/KCNE3 structure, they found that KCNE1 and KCNE3 share a very similar binding site. But different interactions of FTL in KCNE1 and TVD in KCNE3 with KCNQ1 are proposed to account for the different functional modulation of KCNQ1 by different KCNE subunits.

The reviewers agreed that the models deposited in PDB-dev would be of great value to the community, and that the review of experimental restraints used to build the models are in and of itself of great value. They also judged that the experimental validation of the proposed models was thorough and increased the confidence in the proposed modes of interaction between the two proteins. The data were thus convincing, but the conclusions about interactions were sometimes overstated and a more extensive discussion about the mechanism of KCNE1 was needed. Additionally there were some concerns about the way the cross-linking experiments were carried out. A detailed review is included below.

Experiments:

1) Related to Figure 2 and all other experiments using crosslinking: the authors show that under control, reducing or oxidizing conditions the magnitude of the currents are different. The way the data are presented and described led us to understand that each treatment, control, DTT or Cu-Phenantroline was applied to different groups of cells. If this is the case, this is a major experimental problem. Since there is variability in expression level and possibly changes in open probability and/or single channel conductance in each of the mutants that the authors are not accounting for, a simple comparison of the current density magnitude between different treatments is meaningless. These experiments should be carried out in the same cell. Typically disulfide bond formation (crosslinking) is induced by Cu-Phe after basal current recording and then reversed by DTT. There is a vast literature making use of this technique and this is the accepted way of performing this experiment. The authors should clarify if this is indeed what they did and show an example of such an experiment.

Conversely, if each treatment was applied to different groups of cells, the authors should show that the introduced cysteine residue produces no changes in open probability and single-channel conductance.

2) The recordings shown by the authors were obtained in a limited voltage range, up to 60 mV. The tail currents show that the activation is far from saturation at this voltage and in consequence the estimation of the V1/2 from these curves incurs in an error, which could be very large. The authors should comment on these limitations or alternatively estimate this parameter from currents obtained at a wider range of voltages.

Modeling:

1) To strengthen confidence in the modeling procedure, the following controls should be carried out: the docking of E1 should be done on the recent human structures of the activated/open state hKCNQ1 instead of only on the author's previous model. Conversely, docking of E3 to their AO model (and to the experimental structure mentioned above) should be performed to demonstrate that the procedure allows to recapitulate the experimental binding pose.

2) Figure 3—figure supplement 1 shows a comparison of the models obtained in this work with the cryo-EM structure. The authors show that the backbone RMSD is fairly low between models and structure, but it would be more informative to show the degree of deviation of the position of side chains, especially in the regions in the models that were used for docking the KCNE1.

3) Another aspect that is not even mentioned by the authors is the fact that the stoichiometry of KCNQ1-KCNE1 complexes appears to be variable. There is no experimental consensus on how many KCNE1 subunits are present per tetrameric KCNQ1 and the evidence seems to support the view that the number of KCNE1 subunits associated with a KCNQ1 tetramer depends on the expression level of the former. The authors seem to have chosen a 2:1 KCNE1:tetramer assembly for their simulations. Although a different stoichiometry might or might not alter the details of the interaction, it has been show that the variable stoichiometry can alter the biophysical effect (degree of voltage shift among others) of KCNE1, indeed suggesting that the mode of interaction might also not be unique. Did you try docking four KCNE1s per tetramer? Would this different stoichiometry alter the mode of interaction between KCNE1s and KCNQ1? At the very least, the authors should discuss this variable affects and their choice of stoichiometry.

Contextualization and Interpretation of data:

1) Abstract: "FTL… affects the channel gate via an allosteric network." This is not shown, but only a proposed mechanism. Please state as suggestion, not fact.

2) “ Likewise, mutation of FTL to TVG renders the KCNQ1-KCNE1 channel constitutively active, similar to KCNQ1-KCNE3” and other similar statements. Previous studies have shown that KCNQ1/KCNE1-TVG is not a constitutively open KCNQ1/KCNE3-like channel (Panaghie et al., 2006; Barro-Soria et al., 2017), in contrast to the work by Melman. Maybe good to tone down this conclusion? Or more evidence showing that TVG converts KCNQ1-KCNE1 into constitutively conductive KCNQ1-KCNE3 would be good.

3) In Figure 2, why was the focus of the experiments on the extracellular portion of the (TM) region of E1? From the SI tables, there seems to be fewer constraints from interactions of the intracellular portion.

4) “ Both mutations resulted in smaller peak current amplitudes and significantly depolarized activation V_1/2_ (ΔV_1/2_,_K362A_ = 14.7 mV, ΔV_1/2,N365A_ = 14.7 mV) (Figure 5D+E, Supplementary file 1—table 5), indicating that these mutant channels required more energy to open, possibly due to a loss of stabilizing interactions with KCNE1.”. The conclusion is a little too strong. The fact that mutations at K362 and N365 make the channel activate at more depolarized voltages doesn't mean they affect the interaction with KCNE1, even though they seem close to KCNE1 in the model. K362 and N365 are in S6 and maybe just important for pore opening. Crosslinking or double mutant cycle analysis for the C ends of S6 and E1 (say K362 and S68), or KCNE1-dependent effects of K362 and N365 mutations would be needed to claim interactions.

5) Subsection “Experimental validation of KCNE1 residues interacting with KCNQ1” paragraph four. Mutations at W323 and Y46 share similar phenotypes of GV relations, but authors didn't show any experimental evidence to confirm that they are in direct contact (In the model they seem to be close together). Also, Y46 mutant had faster activation kinetics while W323 had wt-like activation kinetics, why is that if they are indeed functionally interacting?

6) Discussion paragraph two. "FTL can affect the nearly PAG motif through the mediation of S5", what residues in S5 is responsible for the mediation? From Figure 9 the authors labeled 263, 266, 267 and 270 and mentioned that mutations of them change the channel activation. Please clarify these mutational effects and discuss how they would differ for the FTL motif and the TVG motif.

7) Discussion paragraph three. This whole paragraph provides some experimental studies to support the model that FTL motif affects PAG motif via S5 involvement. But how this model induces the slower activation gating of KCNQ1 is not clear. To answer this question that the authors mentioned in the beginning of discussion, we would like more evidence and discussion concerning how FTL motif makes the KCNQ1 channel slow to open.

8) Discussion paragraph five. In Barro-Soria et al., 2017, they concluded that FTL also affects the S4 movement.

9) “ This mechanism may account for at least some of the gating differences between KCNE1 and KCNE3” and “ This proposed mechanism can account for the different effects of KCNE1 and KCNE3 on KCNQ1 gating”. This is too strong a statement. Similar to comment 7 above, how does the FTL motif makes Q1 channel open slowly while TVG motif makes Q1 constitutively leaky using this proposed model. It seems like the authors have more ideas about how KCNE1 affects KCNQ1 than how KCNE3 affects KCNQ1? Maybe stick to the more obvious effects seen/proposed?

10) Figure 5. Why does the amplitude go down for mutants? Is this really a good parameter to measure interactions? What is the rationale behind this conclusion?

11) Figure 8. Y267-T58 interaction seems very important for the proposed mechanism, but very little data is shown for this interaction (just simulations, except Y267F reducing expression, but consider concern above). Any more data for this interactions?

12) Figure 7. Y65A looks as good as Y65 or Y65F. So is the aromatic ring at Y65 really important?

13) In several places, data that is used to impose constraints in the model is then used to corroborate the relevance of the model. That should be avoided, or at least clarified.

14) The role of the interaction between Y267 and residues on S4 (M238 and D242) is intriguing, and its disruption when E1 is bound but not E3. Are those state-dependent and can they be related to the stabilization of the specific VSD state?

15) The 274-45 pair that are suggested to crosslink are not that close in either model (15A) and seem to point in different directions.

Presentation of results:

1) In Figure 5, the authors focus on residue W323, Y267 and K362/N365 because they had the largest number of contacts with E1 in a specific segment. Where is the data supporting this claim? The way the contacts are presented in this figure was confusing to several reviewers.

2) Figure 2—figure supplement 2. This is not the correct figure. wt KCNE1 should be shown, not as now with mutant KCNE1. This is a crucial control experiment. Also, the very important controls for wt KCNQ1 coexpressed with KCNE1 cys mutants are missing.

3) Figure 7 B-C. What are the authors trying to convey with these data? The reviewers did not see any obvious pattern. Is the correlation they propose really significant?

---

## [Author Response]

The reviewers agreed that the models deposited in PDB-dev would be of great value to the community, and that the review of experimental restraints used to build the models are in and of itself of great value. They also judged that the experimental validation of the proposed models was thorough and increased the confidence in the proposed modes of interaction between the two proteins. The data were thus convincing, but the conclusions about interactions were sometimes overstated and a more extensive discussion about the mechanism of KCNE1 was needed. Additionally there were some concerns about the way the cross-linking experiments were carried out. A detailed review is included below.Experiments:1) Related to Figure 2 and all other experiments using crosslinking: the authors show that under control, reducing or oxidizing conditions the magnitude of the currents are different. The way the data are presented and described led us to understand that each treatment, control, DTT or Cu-Phenantroline was applied to different groups of cells. If this is the case, this is a major experimental problem. Since there is variability in expression level and possibly changes in open probability and/or single channel conductance in each of the mutants that the authors are not accounting for, a simple comparison of the current density magnitude between different treatments is meaningless. These experiments should be carried out in the same cell. Typically disulfide bond formation (crosslinking) is induced by Cu-Phe after basal current recording and then reversed by DTT. There is a vast literature making use of this technique and this is the accepted way of performing this experiment. The authors should clarify if this is indeed what they did and show an example of such an experiment.Conversely, if each treatment was applied to different groups of cells, the authors should show that the introduced cysteine residue produces no changes in open probability and single-channel conductance.

We agree with the reviewer that ideally the treatments would be carried out on the same cell. However, the I_Ks_ current when analyzed by whole-cell voltage clamp in mammalian cells exhibits rundown (see Vanoye et al., 2018, Supplemental Figure 2), which varies in both rate and magnitude, thus confounding experiments in which there is a time lapse between conditions or treatments. To avoid this problem, we recorded from multiple cells for each treatment (Control, +DTT, +Cu-phen.) to average the cell-to-cell variability. In addition, we recorded cells for all treatments from a specific cell transfection batch, and we used at least 3 transfections for each KCNQ1-KCNE1 combination. This information was previously omitted from Materials and methods, it has now been added to the end of subsection “Electrophysiology”.

We also now show that introduction of cysteine residues to KCNQ1 positions 141 and 274 do not affect channel activity behavior alone or in the presence of wild-type KCNE1 under control, +DTT or +Cu-phen (Figure 2—figure supplement 1 and Figure 2—figure supplement 2, respectively). These results support our conclusion that KCNQ1 residues V141 and I274 are in close proximity to KCNE1 residues L48 and L45, respectively (Figure 2).

2) The recordings shown by the authors were obtained in a limited voltage range, up to 60 mV. The tail currents show that the activation is far from saturation at this voltage and in consequence the estimation of the V1/2 from these curves incurs in an error, which could be very large. The authors should comment on these limitations or alternatively estimate this parameter from currents obtained at a wider range of voltages.

Normalized KCNQ1-KCNE1 tail currents do not saturate at the potentials tested nor would they saturate at more depolarized potentials. This is a widely observed phenomenon for the I_KS_ channel complex and is observed in both mammalian cells and *Xenopus oocytes* (e.g., Wang et al., 2020, Figure 3C; Wang et al., 2012, Figure 3C). We acknowledge this by referring to the V_1/2_ value that we determine by curve fitting as the “apparent” activation V_1/2_. We have changed V_1/2_ to V_1/2app_ throughout the text.

Modeling:1) To strengthen confidence in the modeling procedure, the following controls should be carried out: the docking of E1 should be done on the recent human structures of the activated/open state hKCNQ1 instead of only on the author's previous model. Conversely, docking of E3 to their AO model (and to the experimental structure mentioned above) should be performed to demonstrate that the procedure allows to recapitulate the experimental binding pose.

We thank the reviewers for this suggestion. We have performed these important control experiments and docked the NMR structures of KCNE1 and KCNE3 to the new cryo-EM structure of human KCNQ1 in the AO state and to the Rosetta AO model. The results of these experiments are presented in the paper in Figure 3—figure supplement 4 and in the text:

“To assure the robustness of our structure prediction protocol, control docking calculations were performed with the cryo-EM-determined AO state structure of human KCNQ1 (Sun and MacKinnon, 2020), which became available only after our KCNQ1-KCNE1 models were completed. These control calculations arrived at a model that was very similar to the one developed by docking KCNE1 to the Rosetta homology model of KCNQ1 (interface RMSD (I-RMSD) = 3.5 Å, Figure 3—figure supplement 4A). Additional control calculations were carried out with KCNE3, starting either with the cryo-EM-determined or Rosetta-predicted KCNQ1 model. Guided by a set of published experimental restraints for the KCNQ1-KCNE3 complex (Kroncke et al., 2016), this procedure was able to reproduce the experimental KCNE3 binding pose with an accuracy of I-RMSD = 2.5 Å or 4.2 Å, respectively, for these two structures (Figure 3—figure supplement 4B+C).”

2) Figure 3—figure supplement 1 shows a comparison of the models obtained in this work with the cryo-EM structure. The authors show that the backbone RMSD is fairly low between models and structure, but it would be more informative to show the degree of deviation of the position of side chains, especially in the regions in the models that were used for docking the KCNE1.

We have compared our models of human KCNQ1 used for docking with the cryo-EM structures of the AC and AO state in the region where KCNE1 was docked, i.e. between helices S1+S4 of one subunit and helices S5+S6+P from two neighboring subunits. We have calculated the sidechain RMSD and rotamer similarity of surface-exposed residues in these segments. The results of this analysis have been added to Figure 3—figure supplement 1 and are described in the paper: “In the putative KCNE1 binding region used for docking, the homology models agree well with the cryo-EM structures; surface-exposed residues have a sidechain RMSD less than 2.5Å and 4.0Å in the RC and AO model, respectively (Figure 3—figure supplement 1).”

3) Another aspect that is not even mentioned by the authors is the fact that the stoichiometry of KCNQ1-KCNE1 complexes appears to be variable. There is no experimental consensus on how many KCNE1 subunits are present per tetrameric KCNQ1 and the evidence seems to support the view that the number of KCNE1 subunits associated with a KCNQ1 tetramer depends on the expression level of the former. The authors seem to have chosen a 2:1 KCNE1:tetramer assembly for their simulations. Although a different stoichiometry might or might not alter the details of the interaction, it has been show that the variable stoichiometry can alter the biophysical effect (degree of voltage shift among others) of KCNE1, indeed suggesting that the mode of interaction might also not be unique. Did you try docking four KCNE1s per tetramer? Would this different stoichiometry alter the mode of interaction between KCNE1s and KCNQ1? At the very least, the authors should discuss this variable affects and their choice of stoichiometry.

We thank the reviewers for this thoughtful comment. In addition to describing the preparation of KCNQ1-KCNE1 models with 4:2 stoichiometry in Materials and methods, we have now added more discussion about our choice of the KCNQ1:KCNE1 stoichiometry and the variable effects on KCNQ1 function that can be caused by different numbers of KCNE1 subunits.

“For MD simulations, KCNQ1-KCNE1 models were prepared with a stoichiometry of 4:2 KCNQ1:KCNE1 subunits. […] In our final MD analysis, we focused on the 4:2 stoichiometry and observed no significant changes in the interaction mode between the two KCNE1 subunits and with respect to the model obtained by Rosetta docking.”

Contextualization and Interpretation of data:1) Abstract: "FTL… affects the channel gate via an allosteric network." This is not shown, but only a proposed mechanism. Please state as suggestion, not fact.

We have rephrased this sentence as a suggestion: “The FTL motif binds at a cleft between the voltage-sensing and pore domains and appears to affect the channel gate by an allosteric mechanism.”

2) “ Likewise, mutation of FTL to TVG renders the KCNQ1-KCNE1 channel constitutively active, similar to KCNQ1-KCNE3” and other similar statements. Previous studies have shown that KCNQ1/KCNE1-TVG is not a constitutively open KCNQ1/KCNE3-like channel (Panaghie et al., 2006; Barro-Soria et al., 2017), in contrast to the work by Melman. Maybe good to tone down this conclusion? Or more evidence showing that TVG converts KCNQ1-KCNE1 into constitutively conductive KCNQ1-KCNE3 would be good.

We have corrected these statements to clarify that TVG does not completely convert KCNQ1-KCNE1 into a constitutively open channel.

“Mutation of FTL to TVG renders the KCNQ1-KCNE1 channel similar to KCNQ1-KCNE3, in that faster activation at more negative potentials is observed (Melman et al., 2001; Barro-Soria et al., 2017).”

“Replacement of FTL with TVG from KCNE3 shifts the G(V) curve of KCNQ1-KCNE1 channels towards that of KCNQ1-KCNE3 and removes KCNE1-specific effects on the gate and S4 movement (Barro-Soria et al., 2017).”

3) In Figure 2, why was the focus of the experiments on the extracellular portion of the (TM) region of E1? From the SI tables, there seems to be fewer constraints from interactions of the intracellular portion.

The previous KCNQ1-KCNE1 model by Kang et al., 2008, suggested that the extracellular region of the KCNE1 TMD is in close distance to two gain-of-function mutation sites, V141 and I274, in KCNQ1, and we pursued to test this hypothesis. It has also been our experience that crosslinking of intracellular sites of KCNQ1+KCNE1 expressed in mammalian cells is technically challenging due to the limited accessibility of these sites to the oxidizing/reducing reagents.

We have now added an additional paragraph to the manuscript comparing our models with published interaction data for the intracellular C-terminal domains of KCNQ1 and KCNE1 (see response to the next comment). These restraints were initially not used in model building because the interaction sites are located in flexible linker regions of KCNE1 and inclusion of flexible linkers in docking proved computationally intractable. However, the experimentally observed interaction sites are close to the last KCNE1 residue in our models. This supports the model-predicted proximity between the intracellular portions of KCNE1 and KCNQ1 S6.

4) “ Both mutations resulted in smaller peak current amplitudes and significantly depolarized activation V_1/2_ (ΔV_1/2_,_K362A_ = 14.7 mV, ΔV_1/2,N365A_ = 14.7 mV) (Figure 5D+E, Supplementary file 1—table 5), indicating that these mutant channels required more energy to open, possibly due to a loss of stabilizing interactions with KCNE1.”. The conclusion is a little too strong. The fact that mutations at K362 and N365 make the channel activate at more depolarized voltages doesn't mean they affect the interaction with KCNE1, even though they seem close to KCNE1 in the model. K362 and N365 are in S6 and maybe just important for pore opening. Crosslinking or double mutant cycle analysis for the C ends of S6 and E1 (say K362 and S68), or KCNE1-dependent effects of K362 and N365 mutations would be needed to claim interactions.

We have removed statements from the manuscript which suggested that the effects seen with mutations at K362 and N365 could indicate that these residues interact with KCNE1. Instead we have added a paragraph comparing the structural models with published experimental interaction data for site H363.

“A proximity between the C-terminal ends of the KCNE1 TMD and S6 is supported by the results of cysteine-crosslinking experiments (Lvov et al., 2010), which showed that H363C in KCNQ1 formed disulfide bonds with H73C, S74C, and D76C in KCNE1. […] This result can also be explained by our structural models (Figure 5—figure supplement 3).”

5) Subsection “Experimental validation of KCNE1 residues interacting with KCNQ1” paragraph four. Mutations at W323 and Y46 share similar phenotypes of GV relations, but authors didn't show any experimental evidence to confirm that they are in direct contact (In the model they seem to be close together). Also, Y46 mutant had faster activation kinetics while W323 had wt-like activation kinetics, why is that if they are indeed functionally interacting?

To test if W323 and Y46 are functionally interacting, we have designed a double mutant cycle experiment to measure the free energy changes of channel activation (Δ*G*) for KCNQ1-KCNE1 channels carrying Ala mutations at either KCNQ1 W323, KCNE1 Y46, or at both sites simultaneously. Unfortunately, channels formed by KCNQ1 W323A and KCNE1 Y46A generated very low currents (see Author response image 1), which prohibited determination of Δ*G* for this channel and calculation of the coupling energy between W323 and Y46 (paragraph three subsection “Experimental validation of KCNE1 residues interacting with KCNQ1”).

**Author response image 1. sa2fig1:** Whole-cell currents recorded from CHO-K1 cells expressing wildtype KCNQ1+KCNE1 (left) or KCNQ1 W323A+KCNE1 Y46A (right).

Nevertheless, the prediction of our model that W323 and Y46 are interacting is consistent with the observation that mutations at both sites show similar G(V) relations and that the homologous residue in KCNE3 (Y60) forms a direct contact with W323 in the KCNQ1-KCNE3 channel structure. Furthermore, we observed that mutations at both positions, W323 and Y46, led to faster activation compared to the WT KCNQ1-KCNE1 channel (see Author response image 2), an effect that was more pronounced for mutations to smaller amino acid (Ala, Leu) than for mutations to Phe.

For easier comparison of the kinetics of channels formed by different KCNQ1 and KCNE1 mutants in this work, we have now plotted the activation and deactivation time constants and show these results in Figure 7 and Figure 5—figure supplement 2 in the manuscript.

**Author response image 2. sa2fig2:** Activation time constants of channels formed with (A) KCNQ1 WT and KCNE1 Y46 mutants and (B) KCNQ1 W323 mutants and KCNE1 WT. (mean ± SEM; * P < 0.001, Student’s t-test).

6) Discussion paragraph two. "FTL can affect the nearly PAG motif through the mediation of S5", what residues in S5 is responsible for the mediation? From Figure 9 the authors labeled 263, 266, 267 and 270 and mentioned that mutations of them change the channel activation. Please clarify these mutational effects and discuss how they would differ for the FTL motif and the TVG motif.

We have clarified the mutational effects seen for residues in S5, which suggest an interaction with KCNE1 FTL.

“Alanine mutational scanning of S5 previously showed that the V_1/2_ shift of KCNQ1 activation by KCNE1 is reduced by mutations Y267A and F270A, and to a smaller extent by L271A, I274A, and F275A (Strutz-Seebohm el al., 2011). […] These data are consistent with a binding of FTL to this region on S5 leading to triggering of changes in the activation gate of KCNQ1.”

We describe differences between the interaction modes of FTL and TVG, and, using Rosetta energy calculations and *in-silico* alanine mutagenesis, compare possible changes in their binding strength with S5.

“We observed different interactions are made by the KCNE1 FTL versus the KCNE3 TVG (Figure 9B+C, Video 4). […] Scoring with the Rosetta energy function suggests a larger binding free energy for FTL compared to TVG (Figure 9D, Figure 9—figure supplement 1C) and a decrease of binding by Ala mutations of residues in S5 (Figure 9—figure supplement 1D). […]”

7) Discussion paragraph three. This whole paragraph provides some experimental studies to support the model that FTL motif affects PAG motif via S5 involvement. But how this model induces the slower activation gating of KCNQ1 is not clear. To answer this question that the authors mentioned in the beginning of discussion, we would like more evidence and discussion concerning how FTL motif makes the KCNQ1 channel slow to open.

We agree with the reviewer that our structural models cannot fully explain how the KCNE1 FTL motif causes slow activation gating of KCNQ1.Therefore, we have revised our discussion and focus on questions which are answered more directly by the models developed in this work.

“How does the KCNE1 FTL motif interact with KCNQ1 to control KCNQ1 activation gating? And, what can be concluded about the TMD binding mode for both KCNE1 and KCNE3 and the mechanism underlying the different impact on channel activation by KCNE3?”

As in our previous manuscript version, we describe the basis for an allosteric mechanism by which KCNE1 appears to affect the KCNQ1 gate, and provide experimental evidences for this mechanism.

We then discuss how differences between the binding modes of KCNE1 FTL and KCNE3 TVG lead to different molecular forces acting on S5, and propose that different force transductions determine the effect of the KCNE proteins on the KCNQ1 gate via involvement of S5.

“Comparison of the binding modes of KCNE1 and KCNE3 also suggests how the activation motifs of these proteins could trigger different effects on the KCNQ1 gate through mediation of S5. […] Together these observations support the notion that FTL and TVG induce distinct interactions of varying strength with S5 that determine the effect of the KCNE subunits on the S6 gate.”

As in our previous manuscript submission, we conclude the discussion with thoughts about whether the TMD binding mode seen for KCNE1 and KCNE3 could be conserved in other KCNE proteins.

8) Discussion paragraph five. In Barro-Soria et al., 2017, they concluded that FTL also affects the S4 movement.

We have corrected this statement: “The FTL and TVG motifs were found to determine whether the KCNE subunits affect the gate and the second S4 movement that is seen in voltage clamp fluorometry studies of KCNQ1-KCNE1 and correlates with the opening of KCNQ1-KCNE1 channels.”

9) “ This mechanism may account for at least some of the gating differences between KCNE1 and KCNE3” and “ This proposed mechanism can account for the different effects of KCNE1 and KCNE3 on KCNQ1 gating”. This is too strong a statement. Similar to comment 7 above, how does the FTL motif makes Q1 channel open slowly while TVG motif makes Q1 constitutively leaky using this proposed model. It seems like the authors have more ideas about how KCNE1 affects KCNQ1 than how KCNE3 affects KCNQ1? Maybe stick to the more obvious effects seen/proposed?

As suggested by the reviewer, we have revised our discussion and stick to the more obvious effects proposed by the structural models (see our response to comment 7). We have adjusted our claims and removed the first statement and rephrased the second statement as a suggestion: “Differences in their interactions with S5 appear to explain the different effects of KCNE1 and KCNE3 on the gate in KCNQ1.”

10) Figure 5. Why does the amplitude go down for mutants? Is this really a good parameter to measure interactions? What is the rationale behind this conclusion?

We agree with the reviewers that current amplitude is not a good parameter to measure KCNQ1-KCNE1 interaction. In fact, in our manuscript, conclusions about the KCNQ1-KCNE1 interaction were not derived from peak current data. We reported these data in order to fully describe the functional phenotype of the KCNQ1- KCNE1 mutants tested. To avoid confusion, we have removed the unnormalized I-V curves from Figure 5E and Figure 7A. Instead, we show the activation and deactivation times for KCNE1 Y46, F57, and Y65 mutants in Figure 7A. We include the current density data in Supplementary file 1—table 6, but have removed statements comparing current density between WT and mutant conditions from the manuscript text.

11) Figure 8. Y267-T58 interaction seems very important for the proposed mechanism, but very little data is shown for this interaction (just simulations, except Y267F reducing expression, but consider concern above). Any more data for this interactions?

To test if KCNQ1 Y267 and KCNE1 T58 are interacting, we have performed a double mutant cycle experiment by substituting Y267 with Phe and T58 with Val, either separately or in combination, and determined the changes in the free energy of activation of the resulting channel complexes.

“The energy changes for KCNQ1 Y267F–KCNE1 (Δ*G* = 0.66 kcal/mol), KCNQ1–KCNE1 T58V (Δ*G* = -0.36 kcal/mol) and KCNQ1 Y267F–KCNE1 T58V (Δ*G* = 0.90 kcal/mol) were not additive, however, the net energy change (|ΔΔG| = 0.60 kcal/mol) was smaller than 1.0 kcal/mol, which is commonly used as lower cutoff to identify two residues as interacting. Thus, we were not able to experimentally confirm an interaction between Y267 and T58. However, we note that the free energy changes at T58 were previously observed to have a pronounced sidechain volume dependency (Strutz-Seebohm el al., 2011) and that substitutions to amino acids involving a more drastic change in sidechain size could reveal a stronger energetic coupling between Y267 and T58 than determined in this work.”

12) Figure 7. Y65A looks as good as Y65 or Y65F. So is the aromatic ring at Y65 really important?

Our data suggest that both sidechain size and OH functionality at KCNE1 Y65 contribute to in the interaction with KCNQ1, because mutations to Phe, which preserves the aromatic ring but lacks the OH group, as well as substitutions with Leu or Ala led to significant changes in channel activation properties as described:

“Mutations at Y65 led to a shift of V_1/2app_ to more positive voltages (ΔV_1/2app,Y65A_ = 4.3 mV, ΔV_1/2app,Y65L_ = 12.3 mV, ΔV_1/2app,Y65F_ = 9.4 mV), and Y65L and Y65A showed faster activation and deactivation kinetics (Figure 7A, right panel). This is consistent with the observation of Y65 forming sidechain packing and hydrogen bond interactions with multiple KCNQ1 residues in the MD simulations.”

13) In several places, data that is used to impose constraints in the model is then used to corroborate the relevance of the model. That should be avoided, or at least clarified.

We apologize for this mistake. We have removed the sentence about mutation W323C from the manuscript. Our intention was to compare the volume effects of different mutations at position 323. We now compare our results to mutational effects at other residues in S6 and helix P that are close to W323.

“Mutations of other residues in S6 (V324, V334) and in the nearby P helix (A300, V307) also caused channel opening at more negative voltages, likely via destabilization of the closed state. This region has been implicated with the positive G(V) shift by KCNE1 (Kakajo et al., 2011).”

14) The role of the interaction between Y267 and residues on S4 (M238 and D242) is intriguing, and its disruption when E1 is bound but not E3. Are those state-dependent and can they be related to the stabilization of the specific VSD state?

In our structural modeling work of the KCNQ1 VSD in different activation states we have observed that Y267 can make interactions with different S4 residues in the resting, intermediate, and activated state, respectively (see Author response image 3). It is possible that the type and strength of these state-dependent interactions are affected differently by the presence of KCNE1 or KCNE3 as basis for stabilizing specific VSD states. To fully and quantitatively investigate this hypothesis, we are considering additional computational and functional experiments.

**Author response image 3. sa2fig3:** Interactions between Y267 and S4 in KCNQ1 models of the resting, intermediate, and activated state.

15) The 274-45 pair that are suggested to crosslink are not that close in either model (15A) and seem to point in different directions.

In the model of the KCNQ1-KCNE1 closed state, which appears to be the state that is promoted by I274C-L45C crosslinking, L45 and I274 are oriented at an angle of about 90° and have a CA-CA distance of ~15 Å (see Author response image 4). In MD simulations, which allowed for exploration of additional protein flexibility, L45 and I274 came within a CA-CA distance of 13.5 Å which is still larger than the expected maximal CA-CA cysteine-crosslinking distance of ~8 Å. We speculate that this experimentally observed crosslink may have induced a moderate perturbation of the KCNQ1-KCNE1 structure which brought L45C and I274C closer together than in the WT model inducing a low-conductance closed-like state.

We have added a sentence to the manuscript providing a possible explanation for this restraint violation:

“The restraint with the medium violation involved residues KCNE1 L45 and KCNQ1 I274. It is possible that crosslinking between these cysteine-substituted sites slightly perturbed the KCNQ1-KCNE1 structure leading to a low-conductance closed-like state, which could explain why in the WT channel model these residue sidechains have suboptimal geometry for disulfide bond formation.”

**Author response image 4. sa2fig4:** 

Presentation of results:1) In Figure 5, the authors focus on residue W323, Y267 and K362/N365 because they had the largest number of contacts with E1 in a specific segment. Where is the data supporting this claim? The way the contacts are presented in this figure was confusing to several reviewers.

We apologize for the confusion and have refined Figure 5. The number of contacts that is made by a KCNQ1 residue with KCNE1 is displayed in the surface plots in panels A and B and colored from white (contact number ≤ 1) to red (contact number > 20). For improved readability, we have added the contact numbers in parentheses next to the residue labels.

2) Figure 2—figure supplement 2. This is not the correct figure. wt KCNE1 should be shown, not as now with mutant KCNE1. This is a crucial control experiment. Also, the very important controls for wt KCNQ1 coexpressed with KCNE1 cys mutants are missing.

We apologize for this mistake. When submitting the manuscript, Figure 2—figure supplement 2 got accidently changed to a different figure. We now provide the correct version of Figure 2—figure supplement 2 showing control data for KCNQ1 cysteine mutants co-expressed with KCNE1 WT.

In addition, in Figure 2—figure supplement 3 we show that cysteine-substituted KCNQ1-KCNE1 residue pairs V141C-V47C, I274C-V47C, and I274C-L48C led to no significant changes in peak current in the presence of oxidizing or reducing conditions, respectively. Our negative results with those three KCNQ1-KCNE1 residue pairs indicated to us that the effects we observed in the presence of DTT or Cu-phenanthroline with KCNQ1_V141C-KCNE1_L48C or KCNQ1_I274C-KCNE_L45C are residue pair specific. In addition, earlier experiments carried out in *Xenopus oocytes* (see Author response image 5) showed that the KCNQ1_V141C-KCNE1_L45C pair was insensitive to oxidizing conditions. Together, these results indicated to us that crosslinking was residue-pair specific and we did not test the KCNE1 L45C, V47C and L48C substitutions against wild type KCNQ1.

**Author response image 5. sa2fig5:** Whole-cell currents recorded from oocytes expressing KCNQ1_V141C + KCNE1_L45C treated with control bath solution and Cu-phenanthroline.

3) Figure 7 B-C. What are the authors trying to convey with these data? The reviewers did not see any obvious pattern. Is the correlation they propose really significant?

We validate the model-predicted binding mode for KCNE1 by comparing the location of KCNQ1-contacting residues in KCNE1, indicated in Figure 7B by changes in SASA >20% between KCNE1 alone and KCNE1+KCNQ1, with changes in activation V_1/2_ owing to mutagenesis in KCNE1 displayed in Figure 7C. KCNE1 residues with |ΔV_1/2_| > |ΔV_1/2_|_threshold_ (20 mV for KCNE1 expressed in oocytes in previous studies (Chen et al., 2007; Wang et al., 2012), 10 mV for KCNE1 expressed in CHO-K1 cells in this study) were considered high-impact mutation sites. For ease of comparison, we have refined Figure 7 and highlight high-impact mutation sites and positions with ΔSASA > 20% in blue color. In addition, we have computed the Spearman rank correlation between ΔSASA and |ΔV_1/2_| and performed a χ^2^ test of independence of the observed frequencies of KCNQ1-contact sites and high-impact mutation sites in KCNE1. We compare the significance of this correlation between the model-predicted KCNE1 binding mode and three other hypothetical binding orientations where the KCNE1 TM helix is rotated 90° or 180° in the clockwise or anti-clockwise direction. The results of this analysis are shown in Figure 7—figure supplement 1 and summarized in the text:

“Thus, there is a clear dependence between KCNQ1-contact sites and high impact mutation sites in KCNE1 for the orientation proposed by the structural models (P < 0.05, χ^2^-test). However, no significant correlation was found when we simulated other hypothetical orientations for KCNE1 by rotation around its helical screw axis (Figure 7—figure supplement 1).”